# Geometry-Guided Adversarial Prompt Detection via Curvature and Local Intrinsic Dimension

## Abstract

Adversarial prompts are capable of jailbreaking frontier large language models (LLMs) and inducing undesirable behaviours, posing a significant obstacle to their safe deployment. Current mitigation strategies primarily rely on activating built-in defence mechanisms or fine-tuning LLMs, both of which are computationally expensive and can sacrifice model utility. In contrast, detection-based approaches are more efficient and practical for deployment in real-world applications. However, the fundamental distinctions between adversarial and benign prompts remain poorly understood. In this work, we introduce CurvaLID, a novel defence framework that efficiently detects adversarial prompts by leveraging their geometric properties. It is agnostic to the type of LLM, offering a unified detection framework across diverse adversarial prompts and LLM architectures. CurvaLID builds on the geometric analysis of text prompts to uncover their underlying differences. We theoretically extend the concept of curvature via the Whewell equation into an $n$-dimensional word embedding space, enabling us to quantify local geometric properties, including semantic shifts and curvature in the underlying manifolds. To further enhance our solution, we leverage Local Intrinsic Dimensionality (LID) to capture complementary geometric features of text prompts within adversarial subspaces. Our findings show that adversarial prompts exhibit distinct geometric signatures from benign prompts, enabling CurvaLID to achieve near-perfect classification and outperform state-of-the-art detectors in adversarial prompt detection. CurvaLID provides a reliable and efficient safeguard against malicious queries as a model-agnostic method that generalises across multiple LLMs and attack families.

## 1 Introduction

Frontier Large Language Models (LLMs) are widely used in real-world applications such as education, finance, and legal analysis (Yao et al., 2024). However, adversarial prompts exploit vulnerabilities of LLMs to produce unintended and harmful responses (Wallace et al., 2019; Shen et al., 2023; Deng et al., 2023a). Therefore, ensuring their safety against adversarial prompts is essential to prevent harmful outputs, such as bias, misinformation, or content inciting physical harassment.

Current defences against adversarial prompts rely on prompt engineering or adversarial training. Input perturbation techniques, like Intentionanalysis (Zhang et al., 2024a) and SmoothLLM (Robey et al., 2023), modify the prompt and examine whether the altered version can successfully trigger the LLM's built-in safety mechanisms. The effectiveness of these methods depends on the degree of perturbation and robustness of the LLM's safety alignment. Meanwhile, adversarial training approaches, which fine-tune models to resist adversarial inputs, often struggle to scale to larger LLMs. For example, Latent Adversarial Training (LAT) cannot be easily applied to LLMs exceeding 10 billion parameters (Sheshadri et al., 2024). Given that popular LLMs, such as GPT-3 and PaLM 2, have 175 billion and 340 billion parameters respectively, developing scalable defences for these large models is essential to ensure their reliability and safety (Anil et al., 2023; Brown, 2020).

Existing solutions are inherently tied to the internal architecture and safety alignment training of the targeted LLM, limiting their generality. They do not guarantee consistent performance across different adversarial prompts and varying models (Chao et al., 2023; Shen et al., 2023; Zhou et al., 2024b). In

parallel, Llama Guard Inan et al. (2023) and the constitutional classifier (Sharma et al., 2025) are trained to detect harmful adversarial prompts and block them from reaching the LLM. However, both approaches rely on human annotations to differentiate harmful from benign inputs. More importantly, the underlying distinctions between adversarial and benign prompts remain insufficiently understood, underscoring the need for defence strategies that are both generalizable and theoretically grounded.

In this work, we introduce CurvaLID, an LLM-agnostic framework designed to explore generalizable solutions by uncovering the fundamental geometric differences between adversarial and benign prompts. CurvaLID provides a defence mechanism that operates independently of the internal architecture of LLMs, ensuring its generality across diverse models. It enhances LLM safety by preemptively rejecting adversarial inputs.

First, we introduce PromptLID, a sentence-level Local Intrinsic Dimensionality (LID) measure that effectively captures the geometric properties of text prompts. PromptLID calculates LID using a sentence-level-defined local neighbourhood, unlike traditional approaches (Ma et al., 2018; Yin et al., 2024) that rely on token-level neighbourhoods, which are easily influenced by local noise and often dominated by stop words that provide limited semantic information. Prior work has applied LID to characterise adversarial subspaces in the vision domain (Ma et al., 2018) and to assess the truthfulness of LLM outputs using token-level estimation (Yin et al., 2024), but these methods are not agnostic to specific models or LLM architectures, limiting their generalizability. PromptLID addresses these issues by offering a sentence-level, model-agnostic characterization of prompt-level geometry, enabling more robust adversarial prompt detection.

Second, we develop TextCurv, a theoretical framework for defining curvature in an $n$-dimensional Euclidean space of word embeddings. By extending the curvature concept from the Whewell equation (Whewell, 1849), we prove that the angle between two tangent vectors is equivalent to the difference in their tangential angles. This provides the foundation for analyzing word-level geometry, enabling us to quantify curvature in text embeddings and capture semantic shifts.

Through extensive evaluations, we demonstrate that PromptLID effectively quantifies the high-dimensional local subspaces where adversarial prompts reside, while TextCurv captures curvature at the word level. Together, they complement each other in revealing semantic shifts and localized structural deviations in the text manifold, providing new insights into the fundamental differences between adversarial and benign prompts.

Our main contributions can be summarized as follows:

- We propose CurvaLID, a LLM-agnostic detection framework that leverages geometric distinctions between adversarial and benign prompts. By integrating TextCurv and PromptLID, CurvaLID efficiently and effectively detects adversarial prompts, ensuring safety across various LLMs.
- We provide theoretical insights into the design of two novel geometric measures leveraged by CurvaLID. TextCurv extends the Whewell equation to $n$-dimensional Euclidean space, enabling quantification of semantic shifts through word-level curvature. PromptLID analyses the local intrinsic dimensionality across entire prompts, capturing adversarial subspaces effectively.
- CurvaLID successfully detects about 99% of adversarial prompts, outperforming state-of-the-art defences by over 10% in attack success rate reduction across multiple LLMs and adversarial attacks. It is highly time-efficient, requiring only 0.25 GPU hours of training, whereas existing adversarial training methods have significantly higher computational costs, usually exceeding 100 GPU hours.

## 2 RELATED WORK

This section reviews prior research on adversarial attacks and defence mechanisms for LLMs, highlighting their objectives, underlying logic, and key characteristics.

**Adversarial attacks on LLM.** Adversarial attacks on LLMs involve crafted inputs designed to manipulate models into generating harmful content, like offensive language or dangerous instructions (Zou et al., 2023). These attacks range from a single input (zero-shot) to more complex, continuous dialogue scenarios (multi-shot) (Shen et al., 2023; Dong et al., 2023; Wang et al., 2023). This research focuses on zero-shot text prompt attacks, including techniques like text perturbation that adds gibberish or subtly alters input wording and social-engineered prompts that trick LLMs into harmful behaviour (Zou et al., 2023; Schwinn et al., 2023; Chu et al., 2024).

**Adversarial defences for LLM.** There are three primary defences against adversarial attacks on LLMs: input preprocessing, prompt engineering, and adversarial training. Input preprocessing perturbs inputs to disrupt adversarial prompts but may also affect benign ones, with effectiveness depending on perturbation level and targeted LLM (Cao et al., 2023; Robey et al., 2023; Yung et al., 2024). Prompt engineering augments self-defensive behaviour by adding prompts to expose harmful intent, though performance varies across models (Zhang et al., 2024a; Phute et al., 2023; Zhang et al., 2024b). Finally, adversarial training strengthens the LLM's ability to reject harmful prompts by exposing models to adversarial cases during training (Xu et al., 2024b; Jain et al., 2023). However, this approach requires fine-tuning the LLM, making its success dependent on the specific model being protected, with most adversarial training methods confined to white-box models. Additionally, these methods often demand significant computational resources, with training times reaching up to 128 GPU hours (Mazeika et al., 2024) or 12 GPU hours (Sheshadri et al., 2024), and generally exhibit varying effectiveness across different adversarial prompts and LLMs (Sheshadri et al., 2024; Ziegler et al., 2022; Ganguli et al., 2022). Note that our paper belongs to the field of adversarial prompt detection. Unlike existing methods such as perplexity filtering, which rely on LLMs for next-token probability, our method operates independently of the type of the LLM (Hu et al., 2023).

## 3 BACKGROUND AND TERMINOLOGY

This section provides a brief overview of the mathematical definitions of LID and curvature.

### 3.1 LOCAL INTRINSIC DIMENSION

Local Intrinsic Dimensionality (LID) measures the intrinsic dimensionality of the local neighbourhood around a reference sample (Houle, 2017). Compared to Global Intrinsic Dimension (GID) (Tulchinskii et al., 2024; Pope et al., 2021b), which measures the degree of the $d$-dimension of the global manifold of a data subset, LID focuses on the local neighbourhood of given points. Thus, LID is particularly useful in analyzing high-dimensional data with varying dimensionalities across the dataset.

**Definition 3.1** (Local Intrinsic Dimension (LID)). (Houle, 2017) LID is mathematically defined as:

$$\text{LID}_F(r) = \frac{r \cdot F'(r)}{F(r)}.$$

We are interested in a function $F$ that satisfies the conditions of a cumulative distribution function (CDF) and is continuously differentiable at $r$. The local intrinsic dimension at $x$ is in turn defined as the limit, when the radius $r$ tends to zero:

$$\text{LID}_F^* \triangleq \lim_{r \to 0^+} \text{LID}_F(r).$$

We refer to the LID of a function $F$, or of a point $\mathbf{x}$, whose induced distance distribution has $F$ as its CDF. For simplicity, we use the term 'LID' to refer to the quantity $\text{LID}_F^*$.

$\text{LID}_F^*$ is the theoretical definition, and in practice, has to be estimated (Levina & Bickel, 2004; Tempczyk et al., 2022). Estimation of LID requires a distance measure and a set of reference points to select nearest neighbors (NN). Following prior work (Gong et al., 2019; Ansuini et al., 2019; Pope et al., 2021a; Zhou et al., 2024a; Huang et al., 2024; 2025), we use Euclidean distance. The representation of a data point, along with the chosen reference points, significantly influences how LID is interpreted. In adversarial prompt detection, the representation and neighbourhood definition directly affect the ability to distinguish between clean and adversarial prompts. Among existing estimators, we use the Method of Moments (MoM) (Amsaleg et al., 2015) for its simplicity.

### 3.2 CURVATURE

The intuition of curvature is how quickly a curve changes direction. In geometry, we can visualise curvature through an osculating circle. Curvature can be measured at a given point by fitting a circle to the curve on which the point resides (Kline, 1998). The formal definition is as follows:

**Definition 3.2.** [Curvature measured by osculating circle] (Kline, 1998) The osculating circle at a point $P$ on a curve is the circle tangent at $P$ and passing through nearby points on the curve. Let $R$ be its radius. The curvature $\kappa$ is then defined as:

$$\kappa = \frac{1}{R}.$$

For an arbitrary curve, one can extend the concept of curvature to the rate of tangential angular change with respect to arc length, which is known as the Whewell equation (Whewell, 1849). The tangential angular change refers to the change of angle of inclination of the tangent at the given point.

**Definition 3.3.** [Curvature by Whewell equation](Whewell, 1849) Let $s$ be the arc length and tangential angle $\phi$ be the angle between the tangent to point $P$ and the x-axis, for a given point $P$ on a curve. The curvature $\kappa$ is defined as:

$$\kappa = \frac{d\phi}{ds}.$$

Furthermore, in differential geometry, the curvature can be defined as the change of the unit tangent vector with respect to arc length (Shifrin, 2015; O'Neill, 2006).

**Definition 3.4.** [Curvature in differential geometry](Shifrin, 2015) Suppose curve $\alpha$ is parametrized by arc length $s$ and $\mathbf{T}(s)$ is the unit tangent vector to the curve. We define curvature as

$$\kappa(s) = \|\mathbf{T}'(s)\| = \left\|\frac{d\mathbf{T}}{ds}\right\|.$$

Curvature is also defined and utilized in physics. The Frenet-Serret formulas relate curvature to torsion, tangent, normal, and binormal unit vectors (Frenet, 1852). In the Frenet-Serret formulas, the curvature describes the rotational speed along a curve, which is relevant in kinetics and trajectory applications (Huang et al., 2023). It is also used in autonomous driving, robotics, and quantum computing (Hallgarten et al., 2024; Alsing & Cafaro, 2023; Shabana, 2023).

# 4 GEOMETRIC ANALYSIS AND CURVALID

We aim to develop geometric measures that effectively characterise both benign and adversarial prompts at the prompt and word levels. These measures are then utilized for adversarial prompt detection, formulated as a binary classification task defined as follows.

Let $\mathcal{D} = \{(x^i, y^i)\}_{i=1}^n$ be a labelled dataset comprising $n$ i.i.d. samples $x^i$, where each sample is associated with a label $y^i$. In this context, each input $x^i$ represents a text prompt, with the corresponding label $y^i \in \{0, 1\}$ indicating whether the prompt is benign ($y^i = 0$), or adversarial ($y^i = 1$). Let $\mathcal{M}(x)$ denote the geometric measure applied to a prompt $x$, where $\mathcal{M}(x)$ is composed of two complementary components: the prompt-level measure PromptLID$(x)$ and the word-level measure TextCurv$(x)$. These measures are then utilized within an adversarial prompt detection algorithm, formalized as a classification problem where the objective is to minimize the empirical error between the ground-truth labels and the predictions:

$$\arg\min_\theta \mathbb{E}_{(x,y) \in \mathcal{D}}[\ell(h(\mathcal{M}(x)), y)],$$

where $\ell(\cdot)$ denotes the cross-entropy loss function, and $h(\mathcal{M}(x))$ is the classifier applied to the geometric measures $\mathcal{M}(x) = (\text{PromptLID}(x), \text{TextCurv}(x))$. Alternatively, the defender may use classical outlier detection methods without access to adversarial prompts during training. In both cases, the detector operates on the same geometric measures. Next, we formally define PromptLID and TextCurv, detailing how they explore geometric properties at the prompt and word levels, respectively. Finally, we provide an overview of CurvaLID, our adversarial prompt detection model.

## 4.1 PROMPTLID: LID ESTIMATION AT THE PROMPT-LEVEL

To characterise the prompt-level geometric properties of benign and adversarial prompts, we propose PromptLID, an LID estimation based on prompt representations obtained from a trained CNN. We first train a model $g$ (CNN) to perform a $k$-class classification task, where the goal is to determine which benign dataset a given prompt belongs to. This involves learning a function $g : \mathcal{B} \rightarrow \mathcal{Q}$ to map the input space $\mathcal{B}$ to the label space $\mathcal{Q}$. The label space is defined as $\mathcal{Q} = \{q_1, q_2, \ldots, q_k\}$, where $k$ is the number of types of benign datasets and is equal to the cardinality of $\mathcal{Q}$. Given a benign prompt dataset $\mathcal{B} = \{(b, q)^i\}_{i=1}^n$, where $b$ is the benign prompt and $q$ is its corresponding label, the model learns to classify each prompt into its respective dataset. The objective function used is categorical cross-entropy, as the task involves multi-class classification. The representation $z_1$, derived from the penultimate dense layer, encodes the prompt as a single vector, which is then used to calculate the

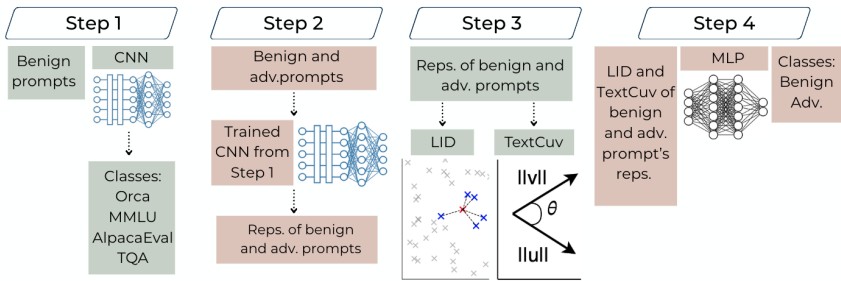

Figure 1: Illustrative diagram of CurvaLID, which classifies benign and adversarial prompts using PromptLID and TextCurv.

PromptLID. The PromptLID expands on the MoM estimation (Amsaleg et al., 2015) of LID on the prompts' representation in $z_1$.

**Definition 4.1.** [PromptLID] The PromptLID of a prompt $x$ is defined as:

$$\text{PromptLID} = -k \cdot \frac{\mu_k}{\mu_k - w^k},$$

where $k$ is the number of nearest neighbors, $\mu_k$ is the mean distance from the prompt representation $z_1$ to its $k$-nearest neighbors, and $w^k$ is the distance to the $k$-th neighbor.

Given that adversarial prompts often manipulate the high-dimensional space of word embeddings to target rarely encountered subspaces (Szegedy, 2013; Ma et al., 2018), they exhibit distinct geometric characteristics. Adversarial prompts are expected to exhibit higher PromptLID as they push the embeddings into regions of the feature space that are less well-defined and more complex than typical benign inputs. As shown in Section 5.2, PromptLID effectively captures this behaviour, highlighting its ability to distinguish between benign and adversarial prompts.

### 4.2  TextCurv: Curvature at the word-level

To characterise word-level geometric properties of benign and adversarial prompts, we analyse the curvature of word connections. The aim is to have curvature complement PromptLID by analyzing the word-level geometric properties of prompts, effectively identifying nearly all adversarial prompts. Specifically, curvature captures the relationships between words, revealing subtle semantic shifts based on word order, uncovering local geometric differences between benign and adversarial prompts.

Prior work shows that CNN activation creates a curved manifold, evidenced by the significantly higher intrinsic dimension estimated by Principal Component analysis (PCA) compared to the GID estimated by TwoNN on activation data (Abdi & Williams, 2010; Facco et al., 2017; Ansuini et al., 2019). However, curvature differences between benign and adversarial prompts remain unexplored. Thus, we examine these differences in convolution layers as potential classification features.

Our goal is to establish a definition of text curvature based on existing mathematical definitions, with the curvature capturing semantic shifts according to word sequence and the strength of these shifts. Word order plays a crucial role in semantic analysis, helping to accurately capture the local geometric properties of prompts. We focus on the representations of prompts in the convolutional layers of the model $g$ as mentioned in Section 4.1, where the prompt data remains unflattened and in stacked lists of vectors at this stage, which can be viewed as word-level representation. Specifically, we extract the representations $z_2$ and $z_3$ from these convolutional layers for further analysis. This stage is critical, as it is where feature spaces are curved, according to prior research (Ansuini et al., 2019).

To capture semantic shifts between consecutive words in a prompt, we draw on Whewell's equation, where the rate of directional change of a curve is represented by the tangential angular change. In NLP, this angular change is connected to the dot-product formula and cosine similarity, which indicate the semantic similarity or difference between two words (Mikolov, 2013; Levy et al., 2015). We assume that this theory also applies to modern word embeddings like GPT-2 and RoBERTa, and therefore, we define the rate of angular change in text curvature accordingly.

**Definition 4.2.** [Text Curvature: Rate of angular change] For any two consecutive word embeddings, denoted by $\vec{u}$ and $\vec{v}$, the rate of angular change, $d\theta$, is defined as:

$$d\theta = \arccos\left(\frac{\vec{u} \cdot \vec{v}}{\|\vec{u}\|\|\vec{v}\|}\right).$$

However, the rate of angular change alone does not fully capture the semantic shift between words, as it overlooks the magnitude of the shift. In differential geometry, curvature is defined by the rate of change in the tangent vector's direction relative to the change in arc length. When two curves exhibit the same directional change, the curve achieving this change over a shorter arc length has a higher curvature. Similarly, in text curvature, given the same semantic shift as measured by our rate of angular change, the curvature should increase when the semantic change is more substantial.

We focus on word vector magnitudes to capture the degree of semantic shift. Previous research suggests that magnitude reflects the semantic weight carried by each word and the tokenizers' understanding of that word within context (Schakel & Wilson, 2015; Reif et al., 2019). For example, common words tend to have smaller magnitudes due to their frequent use and limited semantic significance (Schakel & Wilson, 2015). Instead of summing vector norms to measure distance changes in curvature, which may seem intuitive and consistent with geometric principles, we sum the inverses of the vector norms. This approach is driven by the hypothesis that larger vector norms signify greater semantic importance, meaning that curvature should be inversely proportional to vector norms, capturing larger semantic shifts between words.

**Definition 4.3.** [Text Curvature] For any two consecutive word embeddings, denoted by $\vec{u}$ and $\vec{v}$, the text curvature, denoted by *TextCurv*, is defined as:

$$TextCurv = \frac{d\theta}{\frac{1}{\|\vec{u}\|} + \frac{1}{\|\vec{v}\|}}.$$

The rate of angular change in TextCurv is supported by Theorem 4.4, which links $\theta$ to difference in tangential angles. analysis of how word embedding norms relate to arc length is in Appendix A.2.

**Theorem 4.4.** *For two tangent vectors $\vec{u}$ and $\vec{v}$ in an $n$-dimensional Euclidean space, the angle $\theta$ between them is equivalent to the difference in their tangential angles.*

*Sketch of Proof.* We apply the Gram–Schmidt process to form an orthonormal basis for the tangent space, expressing $\vec{u}$ and $\vec{v}$ as linear combinations of its vectors. In this basis, the angle $\theta$ follows from their inner product. We compute the tangential angles of $\vec{u}$ and $\vec{v}$ from their projections, and by subtracting them show the difference equals $\theta$. The full proof is in Appendix A.2. $\square$

TextCurv captures subtle word-level geometric shifts when adversarial modifications alter a prompt's semantic structure. Adversarial prompts often cause larger and more erratic curvature shifts as they introduce perturbations that disrupt the normal flow of meaning. By analyzing these shifts, TextCurv helps us identify adversarial inputs that deviate from the expected smoothness of benign prompts.

### 4.3 CURVALID: ADVERSARIAL PROMPT CLASSIFICATION BY CURVATURE AND LID

CurvaLID is an adversarial prompt detection method that filters out adversarial prompts before they are input to LLMs, ensuring their safety. Since CurvaLID operates independently of LLMs, it provides a unified defensive performance across all LLMs. This differentiates it from existing SOTA defences like input perturbation, prompt engineering, and adversarial training, which show varying performance across different adversarial prompts and LLMs. Moreover, CurvaLID's evaluation is straightforward and standardised, avoiding the need for subjective human assessments or reliance on LLM judgments, which can raise robustness concerns (Chen et al., 2024; Raina et al., 2024).

CurvaLID involves four steps (see Figure 1, pseudo code in Appendix A.1). In Step 1, we use our trained model $g$ (defined in Section 4.1) to classify different types of benign prompts, deriving the normal feature manifold. This is essential for amplifying the geometric and dimensional distinctions between benign and adversarial prompts. In Step 2, we extract the representations of benign and adversarial prompts from $z_1$, $z_2$, and $z_3$. In Step 3, we compute the PromptLID and TextCurv of each prompt using the representations from Step 2, capturing both sentence-level dimensionality

Table 1: We compare CurvaLID with SOTA defences. The best results are **boldfaced**. Results for other LLMs and defences are provided in Appendix B.4.1.

| LLM | defence | GCG | PAIR | DAN | AmpleGCG | SAP | MathAttack | RandomSearch |
|---|---|---|---|---|---|---|---|---|
| | No defence | 86.0 | 98.0 | 44.5 | 98.0 | 69.0 | 24.0 | 94.0 |
| **Vicuna-7B** | SmoothLLM | 5.5 | 52.0 | 13.0 | 4.2 | 44.6 | 22.0 | 48.5 |
| | Self-Reminder | 9.5 | 48.0 | 35.5 | 11.5 | 25.2 | 22.0 | 6.0 |
| | Intentionanalysis | **0.0** | 8.5 | 3.3 | **0.3** | 0.23 | 20.0 | **0.0** |
| | ICD | 0.2 | 5.2 | 40.4 | 0.9 | 32.8 | 22.0 | 0.2 |
| | RTT3d | 0.2 | 0.3 | 22.0 | 3.5 | 33.5 | 20.2 | 2.5 |
| | **CurvaLID** | **0.0** | **0.0** | **0.0** | 1.1 | **0.0** | **0.0** | **0.0** |
| | No defence | 12.5 | 19.0 | 2.0 | 81.0 | 9.5 | 11.7 | 90.0 |
| **LLaMA2-7B** | SmoothLLM | **0.0** | 11.0 | 0.2 | 0.2 | 1.2 | 11.2 | **0.0** |
| | Self-Reminder | **0.0** | 8.0 | 0.3 | **0.0** | **0.0** | 11.1 | **0.0** |
| | Intentionanalysis | **0.0** | 5.8 | 0.7 | **0.0** | **0.0** | 11.2 | **0.0** |
| | ICD | **0.0** | 2.7 | 0.8 | **0.0** | **0.0** | 10.8 | **0.0** |
| | RTT3d | 0.2 | 0.2 | 1.8 | 0.4 | 5.5 | 9.8 | 0.8 |
| | **CurvaLID** | **0.0** | **0.0** | **0.0** | **0.0** | **0.0** | **0.0** | **0.0** |
| | No defence | 14.9 | 98.0 | 49.7 | 88.9 | 55.1 | 18.9 | 91.9 |
| **PaLM2** | SmoothLLM | 5.5 | 38.7 | 6.7 | 7.2 | 41.2 | 9.8 | 45.3 |
| | Self-Reminder | 2.3 | 36.7 | 22.3 | 4.7 | 21.4 | 13.3 | 3.7 |
| | Intentionanalysis | **0.0** | 2.3 | 1.3 | 0.9 | **0.0** | 9.7 | **0.0** |
| | ICD | 0.1 | 4.9 | 34.2 | 0.2 | 33.9 | 9.3 | **0.0** |
| | RTT3d | 0.1 | 0.1 | 25.5 | 3.3 | 25.0 | 10.2 | 2.8 |
| | **CurvaLID** | **0.0** | **0.0** | **0.0** | **0.0** | **0.0** | **0.0** | **0.0** |

and word-level curvature. In Step 4, we train a Multilayer Perceptron (MLP) to classify benign and adversarial prompts based on the two mean TextCurv values and the PromptLID. The MLP performs binary classification and filters out adversarial prompts before they reach the LLM.

## 5 EXPERIMENTS

In our evaluation, we assessed both the reduction in attack success rate and the prompt classification accuracy of CurvaLID across a range of LLMs (Vicuna-7B-v1.1 (Chiang et al., 2023), LLaMA2-7B-Chat (Touvron et al., 2023), GPT-3.5 (Brown, 2020), PaLM2 (Anil et al., 2023), and Gemma-2-9B (Team et al., 2024)), and compared it against SOTA defences (SmoothLLM (Robey et al., 2023), Self-Reminder (Xie et al., 2023), Intentionanalysis (Zhang et al., 2024a), In-Context Demonstration (ICD) (Wei et al., 2023), RTT3d (Yung et al., 2024), and constrained SFT (Qi et al., 2025)). Our test set has 3,540 prompts, comprising 1,200 benign and 2,340 adversarial prompts. For benign data, we randomly sampled 300 from each of the Orca (Lian et al., 2023), MMLU (Hendrycks et al., 2020), AlpacaEval (Li et al., 2023b), and TruthfulQA (TQA) datasets (Lin et al., 2021). The model $g$ is trained to classify these four benign datasets. For adversarial data, approximately 300 were randomly sampled from each of SAP (Deng et al., 2023a), DAN (also known as the "In The Wild" dataset) (Shen et al., 2023), MathAttack (Zhou et al., 2024b), and GCG (Zou et al., 2023), while around 200 prompts were randomly selected from PAIR (Chao et al., 2023), RandomSearch (Andriushchenko et al., 2024), AmpleGCG (Liao & Sun, 2024), Persuasive Attack (Zeng et al., 2024), AutoDAN (Liu et al., 2023), and DrAttack (Li et al., 2024). Dataset details are in Appendix B.1.5. Unless specified otherwise, we use an 80/20 train–test split. CNN/MLP hyperparameters are in Appendix B.1.2 and B.1.4. Results are averaged over 10 runs for reliability, and ablations are reported in Appendix B.2.

### 5.1 MAIN RESULTS

**Comparison of CurvaLID against baseline defences.** We compared CurvaLID with five existing defences: SmoothLLM, Self-Reminder, Intentionanalysis, In-Context Demonstration (ICD), and RTT3d. Evaluations are conducted across four LLMs: Vicuna-7B-v1.1, LLaMA2-7B-Chat, GPT-3.5, and PaLM2, and against seven adversarial attacks, including GCG, PAIR, DAN, AmpleGCG, SAP, MathAttack, and RandomSearch. In addition, we compared CurvaLID with the constrained SFT defence (Qi et al., 2025), using the fine-tuned Gemma-2-9B model released by Qi et al. (2025). In our approach, if a prompt is classified as a jailbreak prompt, it is rejected. The comparison against SOTA defences across multiple LLMs, measured by ASR (%), is presented in Table 1. Appendix B.4.1

Table 2: Performance metrics for CurvaLID on benign and adversarial datasets. Standard deviations are shown in parentheses.

Table 3: Average TextCurv of benign and adversarial datasets across word embedding and CNN layers. Percentage in parentheses shows the increase in TextCurv for adversarial prompts.

| Class | Accuracy by Dataset | | | | Class Acc. | Overall Acc. | F1 |
|---|---|---|---|---|---|---|---|
| Benign | Orca | MMLU | AlpEval | TQA | 0.984 | | |
| | 0.968 | 1.000 | 0.983 | 0.986 | | 0.992 | 0.992 |
| | (0.012) | (0.000) | (0.008) | (0.010) | (0.009) | | |
| Adv. | SAP | DAN | MathAtk | GCG | 1.000 | | |
| | 1.000 | 1.000 | 1.000 | 1.000 | | | |
| | (0.000) | (0.000) | (0.000) | (0.000) | (0.000) | | |

| Word Embedding | Conv Layer 1 | | Conv Layer 2 | |
|---|---|---|---|---|
| | Benign | Adv. | Benign | Adv. |
| RoBERTa | 0.626 | 0.881 (+40.7%) | 0.325 | 0.446 (+37.2%) |
| GPT-2 | 0.805 | 1.11 (+37.9%) | 0.389 | 0.546 (+40.4%) |
| BERT | 0.428 | 0.590 (+37.9%) | 0.199 | 0.264 (+32.7%) |
| XLNet | 0.296 | 0.431 (+45.6%) | 0.199 | 0.264 (+32.7%) |
| DistilBERT | 0.386 | 0.557 (+44.3%) | 0.225 | 0.322 (+43.1%) |

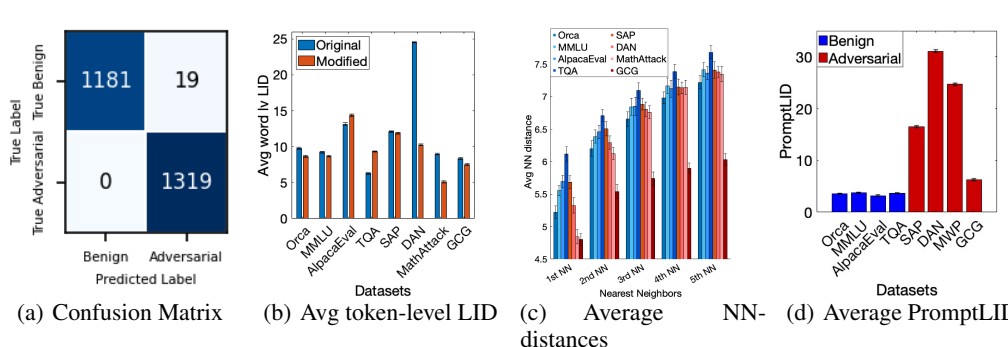

(a) Confusion Matrix   (b) Avg token-level LID   (c) Average NN-distances   (d) Average PromptLID

Figure 2: (a) Confusion matrix from CurvaLID on English adversarial prompts (corresponds to Table 2). (b) Average token-level LID for benign and adversarial prompts. Blue bars show LID for original prompts; red bars show LID after removing stopwords and punctuation. (c) Average nearest neighbor distances, blue for benign and red for adversarial prompts. (d) Average PromptLID for benign and adversarial prompts. Error bars in (b)–(d) show standard deviation over 10 runs.

provides detailed experimental results and covers filters like Circuit Breakers (Zou et al., 2024), Llama Guard (Inan et al., 2023), and perplexity filtering (Alon & Kamfonas, 2023). Experimental result shows that CurvaLID effectively identifies adversarial prompts and rejects them before querying the LLM. Notably, CurvaLID is model-agnostic and performs consistently across LLMs, achieving superior results over baseline defences in most scenarios.

**CurvaLID on adversarial prompts.** The experimental results in Table 2 demonstrate CurvaLID's high performance. The model achieved an overall accuracy of 0.992, with perfect accuracy of 1.00 (i.e., 100%) in identifying adversarial prompts and 0.984 accuracy in identifying benign prompts. Since adversarial prompts are detected before reaching the LLM, 1.00 accuracy implies a 0% attack success rate, effectively nullifying adversarial attempts. The corresponding confusion matrix is shown in Figure 2(a). Notably, CurvaLID remains robust even when the number of prompts per dataset is halved to 150, achieving an accuracy and F1 score of 0.988 (see Appendix B.2.12).

In addition to the four main adversarial datasets, which contain a substantial number of prompts and span a diverse range of attack strategies, we extended our experiments to PAIR, RandomSearch, AmpleGCG, Persuasive Attack, AutoDAN, DrAttack, and persona modulation attack. CurvaLID identified all adversarial attacks with near 0.99 accuracy (see Appendix B.2.1 and B.2.4). It also achieved over 0.9 accuracy on demonstration-based attacks, including In-Context Demonstration (Wei et al., 2023) and cipher-based attacks (Yuan et al., 2023) (see Appendix B.2.2), and 0.994 accuracy on adversarial prompts written in nine non-English languages (see Appendix B.2.3). We also evaluated CurvaLID on benchmarks, namely HarmBench adversarial prompts (Mazeika et al., 2024), achieving near-zero ASR, and over-refusal benchmarks (Cui et al., 2024; Röttger et al., 2023), where CurvaLID reduced harmful prompt acceptance by up to 30% (see Appendix B.2.5- B.2.6).

## 5.2 ANALYSIS ON CURVALID

We analyse CurvaLID under three settings: (1) training only on benign prompts, (2) comparing token-level LID with PromptLID, and (3) testing TextCurv across word embeddings.

**Using CurvaLID in one-class classification problems.** We modified step 4 of CurvaLID by replacing the supervised MLP with unsupervised outlier detection methods such as the local outlier factor (LOF) (Breunig et al., 2000) or isolation forest (Liu et al., 2008), which do not require training on adversarial prompts. Despite the absence of adversarial examples during training, LOF and isolation forest methods achieved comparable detection accuracy of around 0.9. Therefore, our framework is task-independent and can be applied to various problem settings. Full results are in Appendix B.2.13.

**Limitations of token-level LID.** We provide the motivation for PromptLID by examining why traditional token-level LID fails to effectively distinguish benign from adversarial prompts. Adversarial inputs often manipulate rarely encountered regions of the feature space using complex words and irregular combinations (Wallace et al., 2019; Ilyas et al., 2019; Ren et al., 2019). Prior work has shown that such perturbations lead representations into subspaces with distinct local dimensional properties, typically exhibiting high LID (Ma et al., 2018). Based on this, we hypothesize that each word in an adversarial prompt may induce a representation with elevated LID, and that by aggregating these token-level values (e.g., via averaging), it might be possible to detect adversarial prompts.

However, our analysis reveals that this token-level approach, computed with RoBERTa embeddings and treating each word as a data point within its prompt-based neighbourhood, is ineffective at separating benign and adversarial inputs. As shown in Figures 2(b) and Appendix B.3.3, average token-level LID across datasets centers around 10 with high variability, causing substantial overlap between benign and adversarial classes. To investigate further, we analysed the first five nearest-neighbor distances, which also showed minimal differences between prompt types (Figure 2(c)). Appendix B.3.2 indicates the most common neighbors are stop words and punctuation, suggesting token-level LID is dominated by non-informative tokens and insensitive to sequential structure. Removing stop words and punctuation lowered the standard deviation to 2.26 (Appendix B.3.3), but the distinction between benign and adversarial prompts remained weak. These results reaffirm the limitations of token-level LID for adversarial detection and further motivate using PromptLID.

We analyse PromptLID across benign and adversarial datasets. As shown in Figure 2(d), adversarial prompts exhibit a much higher average PromptLID compared to benign prompts, highlighting its effectiveness in distinguishing adversarial prompts. The distribution of PromptLID between benign and adversarial prompts can be found in Appendix B.2.26.

**Generalisation to different word embedding.** We analysed the average TextCurv of adversarial prompts in CurvaLID Step 2. To ensure TextCurv's independence from the embedding model, we conducted curvature analysis using GPT-2 (Radford et al., 2019), BERT (Devlin, 2018), XLNet (Yang, 2019), and DistilBERT (Sanh, 2019). As shown in Table 3, adversarial prompts consistently exhibited at least 30% higher curvature than benign prompts across all embeddings. The TextCurv distribution for benign and adversarial prompts are in Appendix B.2.26. These findings support our hypothesis that words in adversarial prompts exhibit greater irregularity and complexity compared to those in benign prompts. More importantly, the results generalize across different embedding models, suggesting that adversarial and benign prompts differ fundamentally in their geometric properties.

We also demonstrated that CNN activation significantly amplifies TextCurv differences between benign and adversarial prompts. The mean TextCurv of adversarial prompts is at least 30% higher than that of benign prompts in both CNN layers. In contrast, when using only the word embedding, the mean TextCurv is 4.91 for benign prompts and 5.42 for adversarial prompts, a much smaller difference of 13%, less than half of that observed in the CNN layers (see Appendix B.2.25).

## 6 CONCLUSION

In this paper, we introduce CurvaLID, an adversarial prompt detection framework that filters out adversarial prompts before reaching LLMs to maintain their security. CurvaLID operates independently of LLMs, providing consistent performance across models. It achieves over 0.99 accuracy and reduces the ASR of tested adversarial prompts to near zero. CurvaLID leverages PromptLID and TextCurv, which analyse the geometric properties of prompts at the prompt and word level, respectively. These measures address limitations of word-level LID caused by stop words and punctuation, forming the foundation for CurvaLID's robust performance in distinguishing benign and adversarial prompts. Future work includes evaluating CurvaLID on benign prompts in other languages to strengthen multilingual robustness and ensure fairness in low-resource settings.

## REPRODUCIBILITY STATEMENT

The algorithm outline of CurvaLID is presented in Section 4.3, with corresponding pseudo code in Appendix A.1. The definitions and methodologies of PromptLID and TextCurv are described in Sections 4.1 and 4.2, respectively. Theoretical results and proofs are provided in Section 4 and Appendix A.2. Experimental details are given in the first paragraph of Section 5 and Appendix B.1, with dataset descriptions in Appendix B.1.5. Model parameters and architectural configurations are reported in Appendices B.1.2 and B.1.4. The algorithm used in this research is included in the supplementary material, and all data used will be released upon publication.

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

## A   PSEUDOCODE AND THEORETICAL PROOFS FOR CURVALID

This section includes the pseudocode and supplementary theoretical proofs for CurvaLID.

### A.1   PSEUDOCODE FOR CURVALID

Algorithm 1 presents the pseudocode for CurvaLID.

---
**Algorithm 1** CurvaLID

---
**Input:** Datasets $\mathcal{D}_b$, $\mathcal{D}_a$ (benign and adversarial prompts)
**Step 1: Data Preparation**
- Load datasets $\mathcal{D}_b$ and $\mathcal{D}_a$.
- Compute word embeddings $E_b$ and $E_a$ for $\mathcal{D}_b$ and $\mathcal{D}_a$.
**Step 2: Preprocessing**
- Pad sequences to uniform length $L_{\max}$.
- Standardize embeddings to zero mean and unit variance.
**Step 3: Train CNN for Benign Classification**
- Train a CNN $\mathcal{G}$ on $\mathcal{D}_b$ to extract prompt-level representations $z_1$.
**Step 4: Compute PromptLID and TextCurv**
- Calculate PromptLID on $z_1$.
- Extract intermediate layer outputs $z_2$, $z_3$, and calculate TextCurv.
**Step 5: Train the Detection Model**
- Combine PromptLID and TextCurv as features.
- Train an MLP $\mathcal{H}$ for binary classification of benign vs adversarial prompts.

---

### A.2   THEORETICAL FOUNDATIONS AND MATHEMATICAL JUSTIFICATION OF TEXTCURV

We begin by addressing the reference of *TextCurv* to Whewell's equation, specifically focusing on how embedding angles relate to tangential angles. Our objective is to demonstrate that the angle between two word embedding vectors corresponds to the difference in their tangential angles (i.e., the numerator in Whewell's equation). To this end, we prove that for two tangent vectors, $\vec{u}$ and $\vec{v}$, in $n$-dimensional Euclidean space, the angle $\theta$ between them is equivalent to the difference in their tangential angles. The following proof establishes this equivalence, showing that the angular difference between two vectors directly corresponds to the difference in their tangential angles.

**Theorem** For two tangent vectors $\vec{u}$ and $\vec{v}$ in $n$-dimensional Euclidean space, the angle $\theta$ between them is equivalent to the difference in their tangential angles.

**Proof of Theorem**

**Step 1: Angle in $n$-Dimensional Space**

The angle $\theta$ between two vectors $\vec{u}, \vec{v} \in \mathbb{R}^n$ is defined as:

$$\cos\theta = \frac{\vec{u} \cdot \vec{v}}{\|\vec{u}\|\|\vec{v}\|},$$

**Step 2: Tangential Angles and Plane Reduction**

- **Tangential Angles**: Tangential angles describe the orientation of vectors within the specific 2D plane they span. These are defined relative to a chosen reference axis in that plane.

- **Plane Spanned by $\vec{u}$ and $\vec{v}$**: Any two vectors in $n$-dimensional space span a **2D subspace** (a plane). This means the interaction between $\vec{u}$ and $\vec{v}$ (e.g., the angle $\theta$) is fully determined by their projections into this plane.

- **Orthonormal Basis for the Plane**: Using the Gram-Schmidt process, construct an orthonormal basis $\{\vec{e}_1, \vec{e}_2\}$:
  - Normalize $\vec{u}$ to define $\vec{e}_1$:
  $$\vec{e}_1 = \frac{\vec{u}}{\|\vec{u}\|}.$$

- Define $\vec{e}_2$ as orthogonal to $\vec{e}_1$ and lying in the same plane:

$$\vec{e}_2 = \frac{\vec{v} - (\vec{v} \cdot \vec{e}_1)\vec{e}_1}{\|\vec{v} - (\vec{v} \cdot \vec{e}_1)\vec{e}_1\|}.$$

**Step 3: Expressing $\vec{u}$ and $\vec{v}$ in the Orthonormal Basis**

In the orthonormal basis $\{\vec{e}_1, \vec{e}_2\}$:

$$\vec{u} = \|\vec{u}\|\vec{e}_1,$$

and:

$$\vec{v} = a\vec{e}_1 + b\vec{e}_2,$$

where:

$$a = \vec{v} \cdot \vec{e}_1, \quad b = \vec{v} \cdot \vec{e}_2.$$

**Step 4: Computing $\cos\theta$**

The angle $\theta$ between $\vec{u}$ and $\vec{v}$ is:

$$\cos\theta = \frac{\vec{u} \cdot \vec{v}}{\|\vec{u}\|\|\vec{v}\|}.$$

Substituting:

$$\vec{u} \cdot \vec{v} = \|\vec{u}\|a, \quad \|\vec{v}\| = \sqrt{a^2 + b^2}.$$

Thus:

$$\cos\theta = \frac{a}{\sqrt{a^2 + b^2}}.$$

**Step 5: Computing $\sin\theta$**

The magnitude of the cross product $\|\vec{u} \times \vec{v}\|$ in the 2D plane is related to $\sin\theta$ by:

$$\|\vec{u} \times \vec{v}\| = \|\vec{u}\|\|\vec{v}\|\sin\theta.$$

Substituting:

$$\|\vec{u} \times \vec{v}\| = \|\vec{u}\||b|.$$

Thus:

$$\sin\theta = \frac{|b|}{\sqrt{a^2 + b^2}}.$$

**Step 6: Computing $\tan\theta$**

The tangent of $\theta$ is:

$$\tan\theta = \frac{\sin\theta}{\cos\theta}.$$

Substituting:

$$\tan\theta = \frac{\frac{|b|}{\sqrt{a^2+b^2}}}{\frac{a}{\sqrt{a^2+b^2}}}.$$

Simplify:

$$\tan\theta = \frac{|b|}{a}.$$

Thus:

$$\theta = |\arctan(b/a)|.$$

**Step 7: Relating $\theta$ to the Tangential Angles**

In the 2D plane:

- The tangential angle of $\vec{u}$ relative to $\vec{e}_1$ is:

$$\alpha_{\vec{u}} = 0 \quad (\vec{u} \text{ lies entirely along } \vec{e}_1).$$

- The tangential angle of $\vec{v}$ is:

$$\alpha_{\vec{v}} = \arctan(b/a).$$

The difference in tangential angles is:

$$|\alpha_{\vec{u}} - \alpha_{\vec{v}}| = |\arctan(b/a)|.$$

Thus, the geometric angle $\theta$ between $\vec{u}$ and $\vec{v}$ satisfies:

$$\theta = |\alpha_{\vec{u}} - \alpha_{\vec{v}}|.$$

This completes the proof. $\square$

Now we investigate the relationship between word embedding vector norms and the change of arc length. We start with the goal of approximating the arc length $\Delta s$ between two consecutive word embeddings, $\vec{u}$ and $\vec{v}$, in a high-dimensional space. Arc length is classically defined as the integral of the norm of the tangent vector along the curve. For discrete data points, this is approximated as a sum of the Euclidean distances between points.

**1. Discrete Approximation of Arc Length**

Given consecutive embeddings $\vec{u}$ and $\vec{v}$, the arc length between these points can be approximated as:

$$\Delta s = \|\vec{u} - \vec{v}\|.$$

However, directly using $\|\vec{u} - \vec{v}\|$ would treat the embeddings purely as geometric points and ignore their semantic significance as encoded by the vector magnitudes.

**2. Semantic Weight and Embedding Norms**

In NLP, the norm of a word embedding, $\|\vec{u}\|$, encodes the semantic "weight" or importance of a word within its context (Schakel & Wilson, 2015; Reif et al., 2019). Larger norms indicate that the embedding carries more semantic information, while smaller norms suggest less significance.

For two consecutive embeddings, $\vec{u}$ and $\vec{v}$, their combined semantic importance is proportional to their norms:

$$\text{Semantic Importance} \propto \|\vec{u}\| + \|\vec{v}\|.$$

**3. Inverse Proportionality and Arc Length**

To align with the geometric principle in Whewell's equation that relates curvature ($\kappa$) and arc length ($\Delta s$) as:

$$\kappa \propto \frac{1}{\Delta s},$$

we posit that arc length ($\Delta s$) should *decrease* when the semantic importance ($\|\vec{u}\| + \|\vec{v}\|$) increases.

This motivates the choice of the *inverse relationship*:

$$\Delta s \propto \frac{1}{\|\vec{u}\| + \|\vec{v}\|}.$$

**4. Sum of Inverse Norms as Arc Length**

While $\|\vec{u}\| + \|\vec{v}\|$ represents the combined semantic importance of two embeddings, directly using it in the denominator would contradict the inverse proportionality between $\Delta s$ and $\kappa$. Instead, we take the *inverse of the norms individually*, which ensures the arc length is smaller for larger semantic weights.

Thus, the arc length approximation becomes:

$$\Delta s \propto \frac{1}{\|\vec{u}\|} + \frac{1}{\|\vec{v}\|}.$$

The reasoning is that embeddings with larger norms (higher semantic significance) should have smaller contributions to the overall arc length, reflecting the sharper semantic transitions between significant words.

## B  Supplementary information for experiments

The appendix is organized into four sections. B.1 provides supplementary information about CurvaLID. B.2 focuses on ablation studies, presenting experiments to analyse various aspects of CurvaLID's performance. B.3 presents supplementary information on LID analysis. Finally, B.4 evaluates the performance of other SOTA defences, comparing them to CurvaLID. All experiments were conducted on a system with a single Nvidia H100 GPU, 8 CPU cores, and 128 GB of RAM.

### B.1  Supplementary information about CurvaLID

This section includes time and space complexity of CurvaLID, and also the experimental settings for CurvaLID analysis.

#### B.1.1  Time and space complexity of CurvaLID

The time complexity for PromptLID is $\mathcal{O}(np)$, where $n$ is the number of prompts and $p$ is the dimensionality of the prompt embeddings. In our implementation, we use the representation layer of the CNN to obtain the embeddings. For TextCurv, the time complexity is $\mathcal{O}(nmd)$, where $m$ is the number of words in a prompt and $d$ is the word embedding dimensionality. In our case, we use RoBERTa embeddings for word representations. Therefore, the overall time complexity is $\mathcal{O}(n(p + md))$.

The space complexity for PromptLID is $\mathcal{O}(np)$, as we store n prompts, each with $p$ dimensions. For TextCurv, the space complexity is $\mathcal{O}(nz)$, where $z$ represents the dimensionality of the trained CNN layers used in our computations. Consequently, the space complexity is $\mathcal{O}(n(p + z))$.

#### B.1.2  CNN hyperparameter selection

We conducted a preliminary study for the CNN hyperparameter selection to determine the optimal architecture based on overall CurvaLID detection accuracy, training time, and stability. Stability was evaluated by measuring the average CurvaLID accuracy across 10 random seeds. The study involved experimenting with the following parameters:

**Number of convolutional layers:** Tested configurations with 1 to 5 layers.
**Activation functions:** Evaluated ReLU, ELU, tanh, sigmoid, and softplus.
**Kernel sizes:** Tested kernel sizes ranging from 2 to 5.
**Number of parameters:** Adjusted the dense layer sizes to 64, 128, and 256 units.
**Epochs:** Tested training with 10, 20, 30, 40, 50 epochs.
**Batch sizes:** Evaluated sizes of 16, 32, 64, and 128.
**Optimizers:** Compared Adam and SGD.

The experimental results are shown in Table 4. We observe that the parameter tuning of the CNN does not significantly impact the overall detection accuracy, as the accuracy generally fluctuates around 0.98 to 0.99. Notably, the selected CNN extracts features from 1,000 prompts in under 0.5 seconds, demonstrating its efficiency.

#### B.1.3  Detailed parameters and specifics of the CNN architecture for classifying benign prompts in CurvaLID step 1

The CNN architecture consists of an input layer and two 1D convolutional layers. The first Conv1D layer applies 32 filters with a kernel size of 3 and a ReLU activation function, while the second Conv1D layer increases the number of filters to 64, again using a kernel size of 3 and ReLU activation. The output from the convolutional layers is flattened before passing through a fully connected layer with 128 units and ReLU activation. Finally, the network includes an output layer with four units and a softmax activation to classify the input into four distinct categories: Orca, MMLU, AlphEval, and TQA. The model is compiled using the Adam optimizer, categorical cross-entropy loss, and accuracy as the evaluation metric. Training is conducted over 20 epochs with a batch size of 32 and a validation split of 20%.

Table 4: Performance metrics for CNN hyperparameter selection.

| Hyperparameter | Overall Accuracy | Time (min) |
|:---:|:---:|:---:|
| **No. of Conv. Layers** | | |
| 1 | 0.962 | 13.2 |
| 2 | 0.992 | 14.6 |
| 3 | 0.992 | 14.6 |
| 4 | 0.974 | 15.8 |
| 5 | 0.993 | 15.7 |
| **Activation Function** | | |
| ReLU | 0.992 | 14.8 |
| ELU | 0.993 | 15.1 |
| tanh | 0.977 | 15.1 |
| Sigmoid | 0.975 | 14.5 |
| Softplus | 0.991 | 14.8 |
| **Kernel Size** | | |
| 2 | 0.942 | 14.8 |
| 3 | 0.992 | 14.9 |
| 4 | 0.988 | 14.8 |
| 5 | 0.982 | 15.0 |
| **Dense Layer Size** | | |
| 64 | 0.958 | 14.8 |
| 128 | 0.990 | 15.2 |
| 256 | 0.992 | 15.6 |
| **Epochs** | | |
| 10 | 0.943 | 14.2 |
| 20 | 0.991 | 14.9 |
| 30 | 0.989 | 15.3 |
| 40 | 0.990 | 15.5 |
| 50 | 0.988 | 15.3 |
| **Batch Size** | | |
| 16 | 0.990 | 15.6 |
| 32 | 0.991 | 14.8 |
| 64 | 0.988 | 14.7 |
| 128 | 0.982 | 14.2 |
| **Optimizer** | | |
| Adam | 0.990 | 15.2 |
| SGD | 0.981 | 15.8 |

### B.1.4 DETAILED PARAMETERS AND SPECIFICS OF THE MLP ARCHITECTURE FOR CLASSIFYING BENIGN AND ADVERSARIAL PROMPTS IN CURVALID STEP 4

The MLP architecture consists of two fully connected layers and an output layer. The first layer contains 256 neurons with ReLU activation, followed by a batch normalization and dropout layer with a rate of 0.5 to prevent overfitting. A second layer, with 128 neurons and ReLU activation, is followed by another batch normalization and dropout layer. The final output layer uses softmax activation with two units corresponding to the binary classification of benign and adversarial prompts. The model is compiled using the Adam optimizer, a learning rate of 0.001, and categorical cross-entropy as the loss function, and it is trained over 150 epochs with early stopping to prevent overfitting.

### B.1.5 EXPERIMENTAL SETTINGS FOR SECTION 5.1

We use a total of 3,540 testing prompts, comprising 1,200 benign and 2,340 adversarial prompts.

For benign prompts, we randomly sampled 300 prompts from each of Orca, MMLU, AlphacaEval and TruthfulQA.

For adversarial prompts:

- **SAP**: We gathered 320 SAP200 prompts by randomly selecting 40 prompts from each of the 8 adversarial goals.
- **DAN**: We randomly sampled 350 prompts from the adversarial examples uploaded on their GitHub, covering roughly half of their total prompt set.
- **MathAttack**: We used all 300 adversarial prompts provided on their GitHub.
- **GCG**: We followed their default parameter settings—learning rate = 0.01, batch size = 512, top-k = 256, temperature = 1—to generate a universal adversarial suffix. All 349 adversarial behaviours listed in their GitHub were used.
- **PAIR**: We generated 171 adversarial prompts using PAIR. Their implementation targets 50 adversarial goals per LLM, but does not always succeed in producing a prompt for each goal under the default configuration.
- **RandomSearch**: We retrieved prompts directly from their GitHub and randomly selected 200 unique adversarial prompts, as many were duplicates across LLMs.
- **AmpleGCG**: With author permission, we accessed their adversarial prompts and randomly sampled 200 prompts from the set.
- **Persuasive Attack**: We included 150 prompts uploaded by the authors on Hugging Face.
- **AutoDAN**: We included 150 prompts generated per the authors' GitHub instructions for each targeted LLM.
- **DrAttack**: We tested 150 adversarial prompts generated following the configuration provided in their GitHub.

## B.2 ABLATION STUDIES

This section presents ablation studies for CurvaLID, covering its performance across various adversarial prompts, baseline comparisons, and effectiveness under different training conditions and parameter settings. The section is organized as follows:

- **B.2.1 to B.2.7**: Performance of CurvaLID across various adversarial prompts.
- **B.2.8 to B.2.11**: Baseline comparisons against CurvaLID.
- **B.2.12 to B.2.24**: Performance of CurvaLID under different training conditions and parameter settings.
- **B.2.25 to B.2.27**: Ablation studies on the contributions of TextCurv and PromptLID.

### B.2.1 PERFORMANCE METRICS FOR CURVALID IN PAIR, RANDOMSEARCH, AMPLEGCG, PERSUASIVE ATTACK, AUTODAN, DRATTACK

Table 5 shows the performance metrics for CurvaLID in PAIR, RandomSearch, AmpleGCG, Persuasive Attack, AutoDAN, and DrAttack. Note that due to the abundance of each dataset, we are testing these adversarial datasets against benign datasets individually, i.e., four benign datasets against each adversarial dataset.

We obtained 100 prompts from each of the four benign datasets. The six adversarial datasets (PAIR, RandomSearch, AmpleGCG, Persuasive Attack, AutoDAN, and DrAttack) follow the same configuration described in Appendix B.1.5.

Table 5: Performance metrics for CurvaLID in PAIR, RandomSearch, AmpleGCG, Persuasive Attack, AutoDAN, and DrAttack.

| Adv. Dataset | PAIR | RandomSearch | AmpleGCG | Persuasive Attack | AutoDAN | DrAttack | Avg. |
|---|---|---|---|---|---|---|---|
| Benign Acc. | 0.973 | 1 | 0.975 | 0.952 | 0.973 | 0.975 | 0.975 |
| Adv. Acc. | 1 | 1 | 0.976 | 1 | 1 | 1 | 0.996 |
| Overall Acc. | 0.983 | 1 | 0.975 | 0.962 | 0.978 | 0.980 | 0.986 |
| F1 Score | 0.986 | 1 | 0.987 | 0.974 | 0.986 | 0.987 | 0.985 |

### B.2.2 CURVALID ON DEMONSTRATION-BASED ATTACKS

While our focus was on single-shot adversarial prompts, CurvaLID naturally extends to demonstration-based attacks due to its model-agnostic design. Operating independently of the target LLM, it is unaffected by prior demonstrations or context instructions. Any geometric anomaly within the prompt can be detected and filtered before reaching the LLM, effectively mitigating the attack. It also goes the same for multi-turn jailbreaks as CurvaLID can compute the geometric features per turn.

We conducted additional experiments on In-Context Demonstration and cipher-based attacks (Wei et al., 2023; Yuan et al., 2023). For the former, we tested 400 adversarial prompts using the setup from Appendix B.2.1. For the cipher-based attack, we used 400 prompts from the official GitHub repository. The detailed results are shown in Table 6 below:

Table 6: Performance metrics for CurvaLID on in-context demonstration and cipher-based attacks.

| Attack Type | Benign Accuracy | Adversarial Accuracy | Overall Accuracy |
|---|---|---|---|
| In-Context Demonstration Attack | 0.9894 | 0.973 | 0.9812 |
| Cipher-based Attack | 0.9400 | 0.910 | 0.9250 |

### B.2.3 CURVALID ON NON-ENGLISH ADVERSARIAL PROMPTS

Table 7 demonstrates CurvaLID's effectiveness in detecting non-English adversarial prompts by evaluating it on nine languages from the MultiJail dataset (Deng et al., 2023b). We randomly sampled 300 prompts from each of the nine languages—Chinese (zh), Italian (it), Vietnamese (vi), Arabic (ar),

Korean (ko), Thai (th), Bengali (bn), Swahili (sw), and Javanese (jv)—and tested them individually against 400 benign prompts, adjusted to avoid class imbalance. The benign prompts were sampled by gathering 100 entries from each of four different benign datasets. CurvaLID achieved an overall accuracy and F1 score of 0.994, highlighting its robust ability to detect adversarial prompts across a diverse range of languages.

Table 7: Performance metrics for CurvaLID on non-English adversarial datasets.

| Adv. Dataset | zh | it | vi | ar | ko | th | bn | sw | jv | Avg |
|---|---|---|---|---|---|---|---|---|---|---|
| Benign Acc. | 0.975 | 1.000 | 1.000 | 1.000 | 1.000 | 1.000 | 1.000 | 0.975 | 1.000 | 0.994 |
| Adv. Acc. | 1.000 | 0.984 | 0.984 | 1.000 | 1.000 | 1.000 | 0.984 | 1.000 | 0.984 | 0.993 |
| Overall Acc. | 0.988 | 0.994 | 0.994 | 1.000 | 1.000 | 1.000 | 0.994 | 0.988 | 0.994 | 0.994 |
| F1 Score | 0.987 | 0.994 | 0.994 | 1.000 | 1.000 | 1.000 | 0.994 | 0.987 | 0.994 | 0.994 |

### B.2.4 CURVALID ON PERSONA MODULATION

We evaluate CurvaLID against persona modulation attacks, following the setup in Shah et al. (Shah et al., 2023). We generated 200 attack prompts and randomly sampled 200 benign prompts from our benign dataset, as described in Section 5. The experiment setup follows our main evaluation in Section 5.1.

The results, reported in Table 8, show that CurvaLID achieves close to perfect detection accuracy on both benign and persona modulation prompts, demonstrating its robustness.

Table 8: CurvaLID accuracy on benign prompts and persona modulation attacks.

| Method | Benign | Persona Modulation |
|---|---|---|
| CurvaLID | 0.985 | 0.995 |

### B.2.5 CURVALID ON HARMBENCH

We evaluated HarmBench (Mazeika et al., 2024) and tested GCG, AutoDAN, TAP, DirectRequest, and DAN on LLaMA2-7B and Vicuna-7B with 300 prompts each. We also re-evaluated baseline defences (SmoothLLM, Intentionanalysis, Self-Reminder) on these HarmBench prompts.

As shown in Table 9, CurvaLID achieved near-zero attack success rates and matched or outperformed these baselines.

Table 9: ASR (%) of HarmBench attacks under CurvaLID and baseline defences.

| LLM | defence | GCG | AutoDAN | TAP | DirectRequest | DAN | Average |
|---|---|---|---|---|---|---|---|
| | SmoothLLM | 0.5 | 80.33 | **0** | **0** | 0.5 | 16.27 |
| LLaMA-7B | Intentionanalysis | **0** | 0.33 | **0** | **0** | 1.67 | 0.40 |
| | Self-Reminder | **0** | 0.33 | **0** | **0** | 2.5 | 0.57 |
| | **CurvaLID** | 1.33 | **0** | **0** | **0** | **0** | **0.266** |
| | SmoothLLM | 9.67 | 91.5 | **0** | **0** | 19.67 | 24.168 |
| Vicuna-7B | Intentionanalysis | **0.33** | 11.0 | 1.5 | **0** | 9.5 | 4.47 |
| | Self-Reminder | 9.33 | 92.0 | 7.33 | **0** | 57.5 | 33.23 |
| | **CurvaLID** | 2.5 | **3.67** | 2.33 | **0** | **0** | **1.7** |

### B.2.6 EVALUATION ON OR-BENCH AND XSTEST

We evaluated CurvaLID on over-refusal benchmarks, namely OR-Bench and XSTest (Cui et al., 2024; Röttger et al., 2023). We tested on Vicuna-7B and LLaMA-2-7B and measured its effect

on acceptance and rejection rates. We sampled 200 prompts each from the OR-Bench-Hard and OR-Bench-Toxic splits, and similarly from XSTest-Safe and XSTest-Unsafe. As shown in Tables 10 and 11, CurvaLID preserved LLaMA-2-7B's strong rejection behaviour. For Vicuna-7B, CurvaLID reduced harmful acceptance by up to 30%, with only a modest increase (about 10%) in benign rejections, suggesting it can enhance safety without substantially impacting utility.

Table 10: CurvaLID performance on OR-Bench.

| Dataset | Metric | Model | defence | Rate (%) |
|---|---|---|---|---|
| OR-Bench-Hard | Rejection | LLaMA-2-7B | No defence | 85.5 |
| | | | CurvaLID | 91.0 |
| | | Vicuna-7B | No defence | 52.5 |
| | | | CurvaLID | 60.5 |
| OR-Bench-Toxic | Acceptance | LLaMA-2-7B | No defence | 0.5 |
| | | | CurvaLID | 0.0 |
| | | Vicuna-7B | No defence | 33.5 |
| | | | CurvaLID | 1.5 |

Table 11: CurvaLID performance on XSTest.

| Dataset | Metric | Model | defence | Rate (%) |
|---|---|---|---|---|
| XSTest-Safe | Rejection | LLaMA-2-7B | No defence | 52.0 |
| | | | CurvaLID | 59.5 |
| | | Vicuna-7B | No defence | 12.5 |
| | | | CurvaLID | 25.0 |
| XSTest-Unsafe | Acceptance | LLaMA-2-7B | No defence | 0.5 |
| | | | CurvaLID | 0.0 |
| | | Vicuna-7B | No defence | 24.5 |
| | | | CurvaLID | 2.0 |

### B.2.7 EVALUATION OF BASELINE METHODS ON BENIGN PROMPTS

We conducted an evaluation of baseline methods on benign prompts. Specifically, we compared CurvaLID with SmoothLLM, Self-Reminder, Intentionanalysis, ICD, and RTT3d on both Vicuna-7B and LLaMA2-7B. For the benign dataset, we randomly sampled 200 questions from MMLU. We use MMLU because unlike input-perturbation defences such as SmoothLLM and Intentionanalysis, CurvaLID operates as a detection algorithm without modifying the input or the internal mechanisms of the LLM. Thus, MMLU allows for a fair and consistent benchmark across all methods, as it consists of multiple-choice questions with clear-cut right or wrong answers.

To ensure fairness in measurement, if CurvaLID incorrectly flags a benign MMLU prompt as adversarial, we count the resulting LLM output as incorrect. The results are presented in Table 12 below. We observe that CurvaLID has minimal impact on benign performance. However, we also find that most competing defences similarly maintain high accuracy on MMLU, showing no significant degradation in utility.

Table 12: Accuracy on benign MMLU questions under different defence methods. We compare the utility impact of CurvaLID and baseline defences (SmoothLLM, Self-Reminder, Intentionanalysis, ICD, RTT3d) on Vicuna-7B and LLaMA2-7B. Accuracy is measured as the percentage of correctly answered MMLU prompts. "Original" refers to performance without any defence applied.

| Model | Original | CurvaLID | SmoothLLM | Self-Reminder | Intentionanalysis | ICD | RTT3d |
|---|---|---|---|---|---|---|---|
| Vicuna-7B | 46.0 | 48.0 | 40.0 | 46.0 | 50.0 | 48.0 | 39.5 |
| LLaMA-2-7B | 48.5 | 45.5 | 42.5 | 49.0 | 45.0 | 44.5 | 44.5 |

### B.2.8 BASELINE CLASSIFICATION ACCURACY USING ROBERTA EMBEDDINGS AND MLP

Table 13 presents the classification accuracy and performance metrics using RoBERTa embeddings as input to MLP. The MLP consists of a single hidden layer with 128 units, trained for a maximum of

300 iterations. The classification results show a benign class accuracy of 0.893, an adversarial class accuracy of 0.953, and an overall accuracy of 0.923.

Table 13: Classification accuracy and performance metrics using RoBERTa embeddings as input to MLP

| Class | Dataset Accuracy | | | | Class Acc. | Overall Acc. | F1 |
|---|---|---|---|---|---|---|---|
| Benign | Orca | MMLU | AlpEval | TQA | 0.893 | | |
| | 0.872 | 0.899 | 0.893 | 0.905 | | 0.923 | 0.924 |
| Adv. | SAP | DAN | MWP | GCG | 0.953 | | |
| | 0.9400 | 0.962 | 0.885 | 1.000 | | | |

### B.2.9  BASELINE CLASSIFICATION ACCURACY USING CURVALID WITHOUT PROMPTLID AND TEXTCURV

To investigate the importance of the geometric features PromptLID and TextCurv in CurvaLID, we conducted an ablation study by removing Step 3 in CurvaLID (Figure 1). Specifically, we skipped the calculation of PromptLID and TextCurv, and directly fed the CNN representations from Step 2 into the MLP in Step 4 for binary classification. The goal is to assess the standalone performance of the CNN and MLP setup in CurvaLID, serving as a baseline without geometric information.

The results are presented in Table 14. Although the model performs reasonably well, its overall accuracy and F1 score drop by 7% compared to the full CurvaLID with PromptLID and TextCurv. This highlights the critical contribution of the geometric features in capturing topological differences between benign and adversarial prompts and improving classification robustness.

Table 14: Classification accuracy and performance metrics using CurvaLID without PromptLID and TextCurv (i.e., Step 3 removed). CNN representations are directly used as input to the MLP.

| Class | Dataset Accuracy | | | | Class Acc. | Overall Acc. | F1 |
|---|---|---|---|---|---|---|---|
| Benign | Orca | MMLU | AlpEval | TQA | 0.895 | | |
| | 0.920 | 0.850 | 0.900 | 0.910 | | 0.920 | 0.922 |
| Adv. | SAP | DAN | MWP | GCG | 0.947 | | |
| | 1.000 | 0.952 | 0.845 | 0.992 | | | |

### B.2.10  ASR OF B.2.8 AND B.2.9 BASELINE CLASSIFICATIONS

We investigated the two baselines mentioned in B.2.8 and B.2.9 against seven types of adversarial prompts, including GCG, PAIR, DAN, AmpleGCG, SAP, MathAttack, and RandomSearch. Here we name the baseline in B.2.8 as "RoBERTa+MLP" and the baseline in B.2.9 as "CNN+MLP". In our approach, if a prompt is classified as a jailbreak prompt, it is rejected. The comparison against baseline defences across multiple LLMs, measured by ASR (%), is presented in Table 15.

The RoBERTa+MLP and CNN+MLP baselines perform poorly, particularly on PAIR, DAN, and MathAttack, where ASR remains high across all LLMs. These results highlight the importance of TextCurv and PromptLID in CurvaLID for enabling robust classification and significantly reducing ASR of adversarial prompts across LLMs.

### B.2.11  EMBEDDING SOURCE ABLATION: CNN AND SBERT

CurvaLID employs a single lightweight CNN to produce both word-level (TextCurv) and sentence-level (PromptLID) representations from one input. To examine sensitivity to the sentence–embedding source, we replaced the CNN-derived sentence embeddings with the widely used SBERT model `all-MiniLM-L6-v2` when computing PromptLID. Table 16 presents the comparison between CNN-based and the pretrained LLM embedding sentence embeddings for PromptLID and CurvaLID. It shows that the pretrained LLM embedding approach yields similar performance to our original CurvaLID configuration. Therefore, we conclude that the pretrained LLM embedding does not offer a performance advantage in this setting.

Table 15: Comparison of CurvaLID with baseline defences in multiple LLMs, measured by ASR (%) on seven adversarial prompt types.

| LLM | defence | GCG | PAIR | DAN | AmpleGCG | SAP | MathAttack | RandomSearch |
|---|---|---|---|---|---|---|---|---|
| | No defence | 86.0 | 98.0 | 44.5 | 98.0 | 69.0 | 24.0 | 94.0 |
| **Vicuna-7B** | RoBERTa + MLP | **0.0** | 12.2 | 3.6 | 2.5 | 4.25 | 11.5 | 0.2 |
| | CNN + MLP | 0.2 | 16.4 | 4.1 | 2.9 | **0.0** | 15.0 | 0.7 |
| | **CurvaLID** | **0.0** | **0.0** | **0.0** | 1.1 | **0.0** | **0.0** | **0.0** |
| | No defence | 12.5 | 19.0 | 2.0 | 81.0 | 9.5 | 11.7 | 90.0 |
| **LLaMA2-7B** | RoBERTa + MLP | **0.0** | 2.9 | **0.0** | **0.0** | 1.1 | 7.8 | **0.0** |
| | CNN + MLP | **0.0** | 5.5 | 3.1 | 0.1 | **0.0** | 9.8 | 0.4 |
| | **CurvaLID** | **0.0** | **0.0** | **0.0** | **0.0** | **0.0** | **0.0** | **0.0** |
| | No defence | 12.0 | 48.0 | 6.33 | 82.0 | 0.9 | 10.5 | 73.0 |
| **GPT-3.5** | RoBERTa + MLP | **0.0** | 0.3 | 1.2 | 0.2 | **0.0** | 5.1 | **0.0** |
| | CNN + MLP | **0.0** | 0.6 | 2.1 | 0.3 | **0.0** | 2.9 | 0.1 |
| | **CurvaLID** | **0.0** | **0.0** | **0.0** | **0.0** | **0.0** | **0.0** | **0.0** |
| | No defence | 14.9 | 98.0 | 49.7 | 88.9 | 55.1 | 18.9 | 91.9 |
| **PaLM2** | RoBERTa + MLP | **0.0** | 18.7 | 3.2 | 0.68 | 5.9 | 11.0 | **0.0** |
| | CNN + MLP | 0.1 | 7.8 | 4.8 | 1.2 | **0.0** | 8.4 | **0.0** |
| | **CurvaLID** | **0.0** | **0.0** | **0.0** | **0.0** | **0.0** | **0.0** | **0.0** |

Table 16: Comparison between CNN-based and pretrained LLM (MiniLM) sentence embeddings for PromptLID and CurvaLID. The first two columns report PromptLID-only performance (sentence reps from CNN or MiniLM). The last two columns report the full CurvaLID with PromptLID computed from the respective embeddings.

| Metric | PromptLID (CNN only) | PromptLID (MiniLM only) | CurvaLID (PromptLID from CNN) | CurvaLID (PromptLID from MiniLM) |
|---|---|---|---|---|
| Benign Accuracy | 0.987 | 0.945 | 0.984 | 0.955 |
| Adversarial Accuracy | 0.932 | 1.000 | 1.000 | 1.000 |
| Overall Accuracy | 0.958 | 0.967 | 0.992 | 0.973 |
| F1 Score | 0.958 | 0.960 | 0.992 | 0.967 |

### B.2.12 ACCURACY OF CURVALID WITH LESS DATA

Table 17 shows the performance of CurvaLID when trained with less data. We training and tested CurvaLID with 150 prompts from each dataset, halving the number of prompts used from the main result. All other parameters remained the same.

We observe that training CurvaLID with less data has a minimal impact on overall detection accuracy, as both the overall accuracy and F1 score only decreased from 0.992 to 0.988. This demonstrates the efficiency of CurvaLID in leveraging geometric features, enabling it to maintain high performance even with limited training data. Such robustness underscores CurvaLID's potential for deployment in scenarios where access to large, labelled datasets is constrained, making it practical for real-world applications with data scarcity.

Table 17: Classification Accuracy and Performance Metrics for CurvaLID on Benign and Adversarial Datasets with 150 Data from Each Dataset.

| Class | Dataset Accuracy | | | | Class Acc. | Overall Acc. | F1 |
|---|---|---|---|---|---|---|---|
| Benign | Orca | MMLU | AlpEval | TQA | 0.992 | | |
| | 0.9565 | 1.000 | 1.000 | 1.000 | | 0.988 | 0.988 |
| Adv. | SAP | DAN | MathAtk | GCG | 0.983 | | |
| | 1.000 | 0.966 | 1.000 | 0.969 | | | |

### B.2.13 PERFORMANCE METRICS FOR CURVALID WITH REPLACING MLP BY LOCAL OUTLIER FACTOR OR ISOLATION FOREST

The parameters of the local outlier factor is as follows: n_neighbors=30, metric='chebyshev', leaf_size=10, and p=1.

For isolation forest, the contamination is set as auto.

We are testing 1900 prompts in total, with 100 prompts randomly sampled from each of SAP, DAN, MathAttack, GCG, PAIR, AmpleGCG and RandomSearch, and 300 prompts from each of Orca, MMLU, AlphacaEval and TruthfulQA. The experimental results are shown in Tables 18 and 19

We observe that the accuracy and F1 score of CurvaLID, when using Local Outlier Factor and Isolation Forest, drop from 0.992 to approximately 0.9. Despite this decrease, it is important to note that this version of CurvaLID operates as a one-class classification model, which inherently simplifies the classification task by focusing on distinguishing a single class. The ability of CurvaLID to maintain a decent performance under these constraints highlights its robustness and adaptability, suggesting its potential for handling future and previously unseen adversarial attacks in dynamic real-world settings.

Table 18: Performance metrics for CurvaLID with replacing MLP by local outlier factor

| Metric | Benign | Adversarial | Overall |
|---|---|---|---|
| Accuracy | 0.953 | 0.840 | 0.909 |
| F1 Score | 0.927 | 0.878 | 0.903 |

Table 19: Performance metrics for CurvaLID with replacing MLP by isolation forest

| Metric | Benign | Adversarial | Overall |
|---|---|---|---|
| Accuracy | 0.910 | 0.902 | 0.907 |
| F1 Score | 0.953 | 0.833 | 0.903 |

### B.2.14 CURVALID WITH PROMPTLID OR TEXTCURV ONLY

We demonstrate that both PromptLID and TextCurv are crucial for achieving optimal performance in CurvaLID. When using only PromptLID as the input feature, the model achieves an accuracy of 0.95. However, combining PromptLID and TextCurv boosts the accuracy to over 0.99, showcasing the complementary nature of these features. This improvement highlights how TextCurv captures additional geometric properties that PromptLID alone cannot, enabling a more comprehensive distinction between benign and adversarial prompts.

Table 20 illustrated the performance of CurvaLID if we only use LID or TextCurv as our features.

Table 20: Ablation study comparing LID and TextCurv for benign and adversarial prompt classification.

| | PromptLID | TextCurv (1st Conv. Layer) | TextCurv (2nd Conv. Layer) | TextCurv (Both Conv. Layers) |
|---|---|---|---|---|
| Benign Acc. | 0.987 | 0.690 | 0.738 | 0.960 |
| Adv. Acc. | 0.932 | 0.833 | 0.809 | 0.884 |
| Overall Acc. | 0.958 | 0.783 | 0.783 | 0.777 |
| F1 Score | 0.958 | 0.781 | 0.782 | 0.776 |

### B.2.15 ACCURACY OF CURVALID IN DIFFERENT EMBEDDINGS

We tested CurvaLID using different word embeddings, including popular ones like GPT-2, BERT, XLNet, and DistilBERT. The experimental results show that CurvaLID performs similarly with around 0.99 overall accuracy, regardless of the word embedding used. Therefore, CurvaLID's classification performance is independent of the specific word embedding used, demonstrating its robustness and adaptability for deployment across different LLMs with varying word embedding representations.

Table 21 shows the accuracy of CurvaLID in different embeddings. The experimental result shows that CurvaLID maintains a high classification accuracy under different word embeddings.

Table 21: Performance of CurvaLID with Different Word Embeddings. The table summarizes the classification accuracy and F1 scores for benign and adversarial prompts using various word embeddings.

| Word Embedding | Benign Prompt Accuracy | Adv. Prompt Accuracy | Overall Accuracy | F1 |
|---|---|---|---|---|
| RoBERTa | 0.984 | 1.000 | 0.992 | 0.992 |
| GPT-2 | 0.973 | 1.000 | 0.986 | 0.986 |
| BERT | 0.987 | 1.000 | 0.994 | 0.993 |
| XLNet | 0.991 | 0.989 | 0.990 | 0.990 |
| DistilBERT | 0.970 | 0.992 | 0.982 | 0.980 |

### B.2.16    REDUCTION IN ASR OF VICUNA-7B-V1.5 AFTER APPLYING CURVALID

We measured the reduction in ASR after CurvaLID identified and filtered out the adversarial attacks in Vicuna-7B-v1.5, using the same settings specified in the respective original adversarial prompt papers. As shown in Table 22, CurvaLID successfully reduced the ASR of most attacks to zero, outperforming the studied SOTA defences. The experimental results demonstrate that CurvaLID is highly applicable to real-world LLMs, effectively safeguarding them by detecting adversarial prompts before they are processed. Moreover, CurvaLID outperforms SOTA defences, further highlighting its effectiveness and reliability.

Table 22: Attack success rates (ASR) in percentage after CurvaLID in vicuna-7b-v1.5.

| | SAP | DAN | MathAttack | GCG | PAIR | RandomSearch | AmpleGCG |
|---|---|---|---|---|---|---|---|
| Vanilla | 69 | 41 | 56 | 95 | 98 | 95 | 97.5 |
| CurvaLID | 0 | 0 | 0 | 0 | 0 | 0 | 2.4 |

### B.2.17    REPLACEMENT OF CNN IN CURVALID STEP 1 WITH TRANSFORMER AND RNN MODELS

We experimented with replacing the CNN architecture (Step 1 of CurvaLID) with Transformer and RNN models. The experimental results are shown in Table 23. While both Transformer and RNN achieved comparable detection accuracy (0.98 versus CNN's 0.992), they required almost two to three times longer training times. Hence, employing CNN in Step 1 of CurvaLID proves to be the optimal choice, maintaining high detection accuracy while ensuring relatively low training time. This demonstrates CurvaLID's computational efficiency and practicality. Details of the Transformer and RNN configurations are provided below.

**Transformer:** The transformer has an input layer, a multi-head attention layer with 4 heads and a key dimension of 64, followed by layer normalization, a dense layer with 128 units, dropout (rate of 0.1), and a final layer normalization. The model then flattens the output, adds another dense layer with 128 units, and concludes with a softmax output layer for classification into 4 classes.

**RNN:** The RNN model begins with an input layer, followed by two stacked LSTM layers with 64 units each (the first LSTM layer returns sequences, while the second does not). After the LSTM layers, there is a dense layer with 128 units and a ReLU activation, followed by a softmax output layer for classification into 4 classes. The model is compiled with the Adam optimizer and categorical cross-entropy loss.

Table 23: Comparison of CurvaLID Architectures

| Metric | CNN (Original Setting) | Transformer | RNN |
|---|---|---|---|
| Detection accuracy | 0.992 | 0.984 | 0.989 |
| Overall training time (min) | 14.58 | 39.01 | 25.10 |

### B.2.18 PERFORMANCE OF CURVALID ON ADVERSARIAL PROMPTS WITH REORDERED WORD SEQUENCES

We utilized GPT-4-o to reorder the words in every sentence of the adversarial prompts while preserving their semantic meaning. The experimental results, presented in Table 24, demonstrate that CurvaLID maintains robust performance, achieving an overall accuracy of 0.984 in detecting these adversarial prompts with altered word order. It is important to note, however, that reordering the words in adversarial prompts may potentially disrupt their effectiveness as attacks, as the content and intent of the original prompts could be compromised.

Table 24: Performance metrics for CurvaLID on adversarial prompts with reordered word sequences.

| Class | Dataset Accuracy | | | | Class Acc. | Overall Acc. | F1 |
|-------|------|------|--------|------|-----------|-------------|------|
| Benign | Orca | MMLU | AlpEval | TQA | 0.981 | | |
| | 0.993 | 0.983 | 0.973 | 0.973 | | 0.984 | 0.984 |
| Adv. | SAP | DAN | MathAtk | GCG | 0.987 | | |
| | 0.983 | 0.963 | 1.000 | 1.000 | | | |

### B.2.19 PERFORMANCE OF CURVALID ON REORDERED PERSUASIVE SOCIAL-ENGINEERED ADVERSARIAL PROMPTS

In this section, we evaluate CurvaLID on PAIR, DAN, and Persuasive attacks, all of which are social-engineered persuasive attacks designed to preserve both semantic meaning and adversarial intent while varying structure (Chao et al., 2023; Shen et al., 2023; Zeng et al., 2024). To introduce linguistic variations, we utilized GPT-4-o to reorder the words in each sentence of the adversarial prompts while maintaining their semantic meaning. The experimental setup remains the same as described in B.2.1, with the addition of 300 DAN prompts. The experimental results, presented in Table 25, show that CurvaLID consistently achieved over 96% detection accuracy across all three attack types and maintained a 0% attack success rate on Vicuna. These findings highlight the robustness of our method, even against sophisticated and linguistically varied prompts specifically crafted to bypass defences. This robustness underscores CurvaLID's potential for deployment in real-world scenarios, where adversarial prompts are likely to exploit linguistic diversity to evade detection.

Table 25: Performance of CurvaLID on reordered social-engineered attacks

| Adversarial attack | PAIR | DAN | Persuasive Attack |
|--------------------|------|------|-------------------|
| Benign accuracy | 0.951 | 0.971 | 0.966 |
| Adversarial accuracy | 0.988 | 1.000 | 0.962 |
| Overall accuracy | 0.962 | 0.983 | 0.964 |
| Attack success rate on Vicuna-7B-v1.5 | 0 | 0 | 0 |

### B.2.20 DIFFERENCES IN PROMPTLID AND TEXTCURV BETWEEN BENIGN AND ADVERSARIAL PROMPTS UNDER LINGUISTIC REORDERING

We conducted an experiment to investigate the differences in PromptLID and TextCurv between benign and adversarial prompts after reordering the adversarial prompts. To introduce linguistic variations, we utilized GPT-4-o to reorder the words in each sentence of the adversarial prompts while preserving their semantic meaning. For the benign prompts, we tested 100 samples each from Orca, MMLU, AlpacaEval, and TQA datasets. Similarly, for the adversarial prompts, we tested 100 samples each from PAIR, DAN, and Persuasive attacks (Chao et al., 2023; Shen et al., 2023; Zeng et al., 2024). Table 26 highlights the geometric differences between benign and adversarial prompts, demonstrating the effectiveness of our method in capturing these variations. Notably, even after linguistic reordering of the prompts, both PromptLID and TextCurv continue to exhibit significant distinctions between benign and adversarial prompts. This ensures that the performance of CurvaLID remains unaffected by such reordering, further underscoring the robustness and reliability of these geometric measures in differentiating adversarial inputs.

Table 26: Geometric differences in TextCurv and PromptLID between benign and adversarial prompts. The percentages in parentheses indicate the relative increase in adversarial prompts compared to benign prompts, calculated as $(\text{Adversarial} - \text{Benign})/\text{Benign} \times 100$.

| Geometric Measures | TextCurv@Conv Layer 1 | | TextCurv@Conv Layer 2 | | PromptLID@Dense Layer | |
|---|---|---|---|---|---|---|
| | Benign | Adversarial | Benign | Adversarial | Benign | Adversarial |
| Average Value | 0.644 | 0.813 (+26.3%) | 0.341 | 0.425 (+24.6%) | 3.546 | 18.223 (+413.8%) |

### B.2.21 CURVALID WITH SEPARATED BENIGN TRAINING AND TESTING DATA

We conducted an ablation study by training CurvaLID on two benign datasets and testing it on the remaining two. Specifically, we trained CurvaLID using only Orca and MMLU as benign data and evaluated it on AlpacaEval and TQA. The results, shown in Table 27, demonstrate an overall detection accuracy of 0.982, just one percentage point lower than when trained on all four benign datasets. These findings indicate that CurvaLID's performance remains robust and is not overly optimistic, even when tested on unseen benign datasets. This suggests that the geometric features captured by PromptLID and TextCurv generalise well across different benign datasets, and the small number of errors largely arises from benign prompts with unfamiliar formatting or structure.

Table 27: CurvaLID with different training and testing benign datasets

| Data class | Accuracy by dataset | | | | Accuracy by class | Overall accuracy | F1 score |
|---|---|---|---|---|---|---|---|
| Benign | AlpacaEval | | TQA | | 0.9412 | | |
| | 0.963 | | 0.9194 | | | 0.982 | 0.98 |
| Adversarial | SAP | DAN | MathAttack | GCG | 1 | | |
| | 1 | 1 | 1 | 1 | | | |

### B.2.22 CURVALID WITH LONG TEXT LENGTH BENIGN PROMPTS

We conducted an additional experiment to evaluate CurvaLID's performance on benign prompts with longer text lengths. Specifically, we calculated the median text length of benign prompts (106 characters in our experiment), removed all benign prompts with fewer than the median, and reevaluated CurvaLID's performance. The results, presented in Table 28 below, show that this adjustment had minimal effect on detection accuracy. The overall accuracy was 0.990, compared to 0.992 when all benign prompts (without filtering by text length) were included. This confirms that text length has minimal impact on CurvaLID's performance, highlighting its robustness and ability to generalize across prompts of varying lengths. Such adaptability makes CurvaLID particularly well-suited for real-world applications, where input lengths can vary significantly.

Table 28: CurvaLID on benign prompts over 106 characters

| Data class | Accuracy by dataset | | | | Accuracy by class | Overall accuracy | F1 score |
|---|---|---|---|---|---|---|---|
| Benign | Orca | MMLU | AlpacaEval | TQA | 0.981 | | |
| | 0.922 | 1.000 | 1.000 | 1.000 | | 0.990 | 0.990 |
| Adversarial | SAP | DAN | MathAttack | GCG | 1.000 | | |
| | 1.000 | 1.000 | 1.000 | 1.000 | | | |

### B.2.23 CURVALID WITH NON-STANDARD BENIGN PROMPTS

We conducted an experiment to evaluate CurvaLID's performance on non-standard benign samples. Specifically, we utilized GPT-4-o to introduce spelling errors by replacing one word in each sentence of all benign prompts with a misspelled variant. The experimental results are presented in Table 29 below.

Our findings reveal that the detection accuracy by dataset exhibited minimal changes, and the overall accuracy remained almost identical to the original experiment , which involved benign prompts without spelling errors. These results demonstrate that introducing spelling errors has a negligible impact on CurvaLID's performance, reaffirming its robustness in handling non-standard text inputs.

Table 29: CurvaLID on benign prompts with spelling errors

| Data class | Accuracy by dataset | | | | Accuracy by class | Overall accuracy | F1 score |
|---|---|---|---|---|---|---|---|
| Benign | Orca | MMLU | AlpacaEval | TQA | 0.985 | | |
| | 0.957 | 0.983 | 1.000 | 1.000 | | 0.990 | 0.990 |
| Adversarial | SAP | DAN | MathAttack | GCG | 0.995 | | |
| | 0.990 | 0.990 | 1.000 | 1.000 | | | |

### B.2.24 CURVALID ON HELD-OUT DATASETS (OUT-OF-DISTRIBUTION DATA) AND PROMPT-LENGTH ROBUSTNESS

We performed two additional robustness evaluations. First, we ran a held-out dataset study across four benign and four adversarial datasets (200 prompts per dataset). For each run, CurvaLID was trained on seven datasets and evaluated on the remaining one. Table 30 reports per-dataset accuracy on the held-out sets, showing consistently strong performance (around 0.9), indicating solid out-of-distribution generalisation. The lowest performing dataset is PAIR (0.875 accuracy). PAIR attacks are social-engineering adversarial prompts with high human readability; unlike gradient-based attacks such as GCG or AmpleGCG, which often introduce unnatural or gibberish token patterns, PAIR prompts are fluent, coherent, and unconstrained by artificial token structures. As a result, they closely resemble natural conversational text rather than template-based jailbreaks, producing less curvature irregularity and weaker geometric signals. This explains the small number of prediction errors on this dataset.

Table 30: CurvaLID detection accuracy and false positive rate (FPR) on held-out datasets for OOD evaluation.

| Dataset | Orca | MMLU | AlpacaEval | TruthfulQA | GCG | PAIR | DAN | AmpleGCG |
|---|---|---|---|---|---|---|---|---|
| Accuracy | 0.955 | 0.940 | 0.945 | 0.995 | 1.000 | 0.875 | 1.000 | 1.000 |
| FPR | 0.045 | 0.060 | 0.055 | 0.005 | 0.000 | 0.125 | 0.000 | 0.000 |

Second, we assessed robustness to prompt length by training on prompts shorter than 106 characters (the median benign length) and testing on longer prompts. As summarized in Table 31, the overall accuracy drops by only 0.05, suggesting minimal sensitivity to prompt length.

Table 31: CurvaLID performance under OOD evaluation on prompt length.

| Model | Benign Acc. | Adv. Acc. | Overall Acc. |
|---|---|---|---|
| CurvaLID trained with shorter prompts | 0.911 | 0.965 | 0.938 |
| CurvaLID | 0.984 | 1.000 | 0.992 |

### B.2.25 AMPLIFICATION OF TEXTCURV DIFFERENCES THROUGH CNN ACTIVATION

We investigated how TextCurv differs between benign and adversarial prompts across CNN layers in CurvaLID. As shown in Table 32, we observe that CNN activation significantly amplifies the curvature gap: the mean TextCurv of adversarial prompts is at least 30% higher than that of benign prompts in both convolutional layers. In contrast, when calculated using only the word embeddings, the difference is notably smaller—4.91 for benign prompts versus 5.42 for adversarial prompts—amounting to a 13% increase, less than half of the gap observed in the CNN layers. All results are averaged over 10 independent runs using different random seeds. The experimental settings remain consistent with those described in Appendix B.1.5.

### B.2.26 ANALYSIS OF PROMPTLID AND TEXTCURV DISTRIBUTIONS

Figures 3, 4, and 5 illustrate the distributions of PromptLID and TextCurv for benign and adversarial prompts. The experiment setting follows B.1.5 and the word embeeding used is RoBERTa. Figure 3 demonstrates that adversarial prompts exhibit a significantly wider range of PromptLID values

Table 32: Mean TextCurv values of benign and adversarial prompts based on word embeddings only and across CNN layer representations in CurvaLID.

| Word Embedding | Embedding Only | | Conv Layer 1 | | Conv Layer 2 | |
|---|---|---|---|---|---|---|
| | Benign | Adv. | Benign | Adv. | Benign | Adv. |
| RoBERTa | 4.91 | 5.42 (+13.0%) | 0.626 | 0.881 (+40.7%) | 0.325 | 0.446 (+37.2%) |

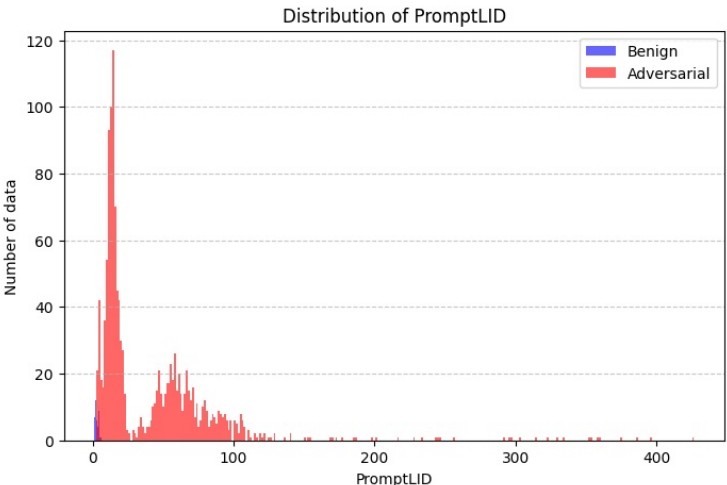

Figure 3: Distribution of PromptLID values for benign (blue) and adversarial (red) prompts, showing the number of data points across different PromptLID ranges.

compared to benign prompts, with higher average values. This suggests that adversarial prompts tend to reside in more complex and sparse regions of the feature space. Figures 4 and 5 display the distributions of TextCurv across the first and second convolution layers, respectively. In both layers, adversarial prompts show consistently higher curvature values, reflecting their tendency to cause greater geometric distortions at the word level. These results highlight the ability of PromptLID and TextCurv to distinguish adversarial prompts based on their unique geometric properties, reinforcing their utility in adversarial prompt detection.

### B.2.27 EFFECTIVENESS OF GLOBAL INTRINSIC DIMENSION IN ADVERSARIAL PROMPT DETECTION.

Global intrinsic dimension (GID) is another plausible approach for the word-level representation. It can avoid aggregating the LID for each word by assessing the GID for each word within the prompt and output a single value. We use the MLE-based estimate from Tulchinskii et al. (Tulchinskii et al., 2024). However, as shown in Figure 6, GID shows no clear distinction between benign and adversarial datasets. Instead, it strongly correlates with prompt length, with Pearson and Spearman correlation coefficients of 0.92 and 0.98, respectively. Even after removing stop words and punctuation, the results were similar, with a Pearson correlation coefficient of 0.9, highlighting the limitations of GID in detecting adversarial prompts.

### B.2.28 ADAPTIVE ATTACK ATTEMPTS AGAINST CURVALID

To evaluate CurvaLID under an adaptive attacker model, we conducted preliminary experiments in which the adversary explicitly attempts to minimise both PromptLID and TextCurv while preserving jailbreak intent. We used a brute-force gradient-based search procedure with the assistance of GPT-4.1 to iteratively propose low PromptLID and TextCurv perturbations that remain semantically adversarial.

In practice, this adaptive objective proved extremely difficult to optimise. Although the minimisation procedure occasionally lowered CurvaLID's confidence scores, the resulting prompts failed to bypass the internal safety mechanisms of the underlying LLMs, meaning the attack no longer achieved its

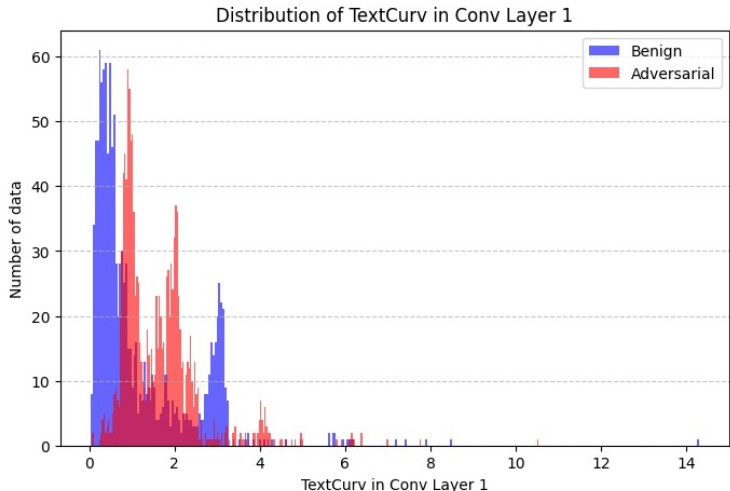

Figure 4: Distribution of TextCurv values in the first convolution layer for benign (blue) and adversarial (red) prompts, indicating the number of data points for each TextCurv range.

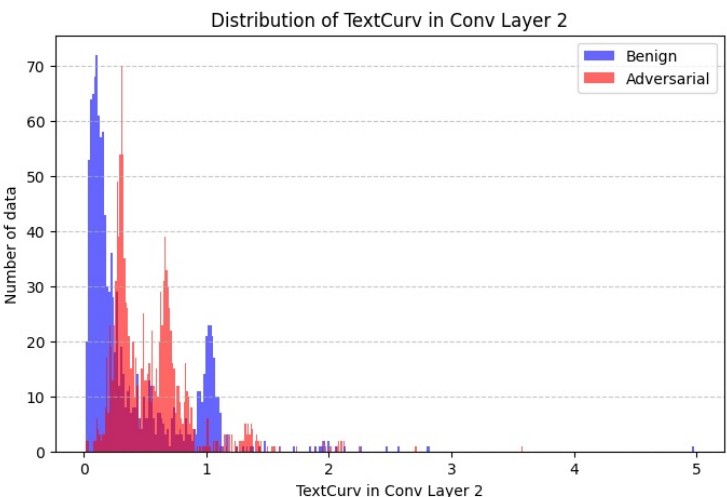

Figure 5: Distribution of TextCurv values in the second convolution layer for benign (blue) and adversarial (red) prompts, indicating the number of data points for each TextCurv range.

primary goal of causing harmful behaviour. In other words, prompts that were successfully optimised to reduce geometric signals systematically lost their jailbreak effectiveness, while prompts that preserved adversarial intent retained high PromptLID and TextCurv values. These findings suggest that simultaneously reducing PromptLID and TextCurv while still executing a successful jailbreak is non-trivial for current attack methods.

### B.2.29 PER-DATASET ROC/AUROC FOR EXPERIMENT IN SECTION 5.1

We compute per-dataset ROC and AUROC scores for each of the four benign and four adversarial datasets used in the main experiment in Section 5.1. As shown in Table 33, CurvaLID achieves uniformly high accuracy and AUROC across all eight datasets, further supporting its robustness and stability.

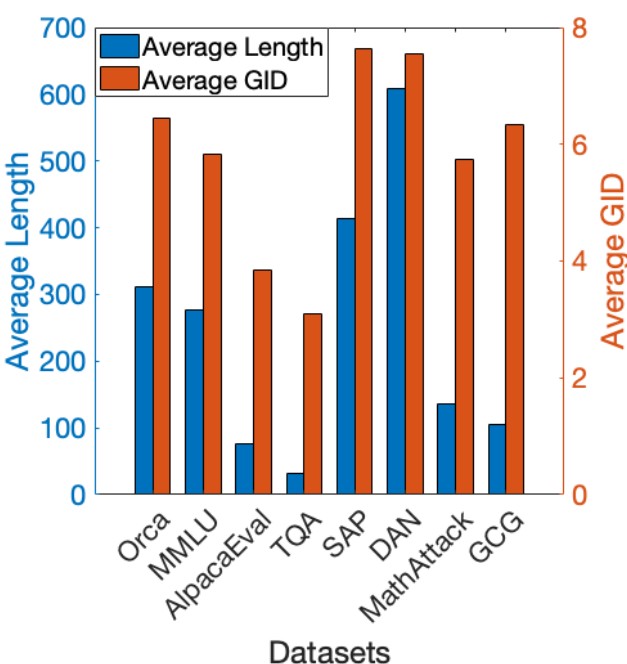

Figure 6: Comparison of average prompt length and global intrinsic dimension (GID) across datasets.

Table 33: Per-dataset accuracy and AUROC for CurvaLID in the main experiment (Section 5.1).

| Dataset | Orca | MMLU | AlpacaEval | TQA | SAP | DAN | MathAtk | GCG |
|---|---|---|---|---|---|---|---|---|
| **Accuracy** | 0.968 | 1.000 | 0.983 | 0.986 | 1.000 | 1.000 | 1.000 | 1.000 |
| **AUROC** | 0.983 | 1.000 | 0.992 | 0.993 | 0.992 | 0.992 | 0.992 | 0.992 |

### B.3 LID ANALYSIS

This section includes supplementary information on LID analysis.

#### B.3.1 LID ESTIMATION USING METHOD OF MOMENTS

This section provides the pseudo code for estimating LID by the Method of Moments, see Algorithm B.3.1. Note that this is different from PromptLID in Definition 4.1.

---

**Algorithm 2** LID Estimation using Method of Moments

---

**Input:** Dataset, Reference points, Number of neighbors $k$
**For each data point in Dataset:**
- Compute pairwise distances $r$ between the data point and all points in Reference.
- Sort distances in ascending order and store them as $a$.
- Compute the mean of the first $k - 1$ nearest distances:

$$m = \frac{1}{k-1} \sum_{i=1}^{k-1} a_i.$$

- Estimate LID for the data point:

$$\text{LID} = \frac{m}{a_k - m}.$$

**Output:** LID values for all data points.

---

#### B.3.2 TOP 10 MOST COMMON NEAREST-NEIGHBORS FOR DIFFERENT PROMPTS IN DIFFERENT DATASETS

Table 34 shows the top 10 most common nearest-neighbors for different prompts in different datasets. The results reveal that the common nearest neighbors in the representation space are predominantly stop words and punctuation. This indicates that word-level LID fails to account for the sequential structure of text and relies on conjunctions, articles, and punctuation. Consequently, these findings highlight the limitations of word-level LID in effectively detecting adversarial prompts.

Table 34: Top 10 most common nearest neighbors for each dataset. The angle brackets (<>) are used to specify punctuation and newline characters in the tokenizer. The visible space symbol (␣) represents a space preceding a word or punctuation.

| Dataset | Top 10 most common nearest neighbors |
|---|---|
| SAP | ␣to, ␣and, ␣the, <.>, ␣a, ␣of, <,>, ␣<">, ␣that, ␣your |
| DAN | <,>, <.>, `<newline>`, ␣the, ␣and, ␣to, ␣you, ␣will, ␣not, ␣is |
| MathAttack | ␣the, <,>, ␣of, ␣he, <.>, ␣to, ␣and, ␣she, ␣is, ␣a |
| GCG | ␣text, tto, use, ized, <}>, ␣mar, dt, ␣a, <'>, ␣Guide |
| Orca | ␣the, <,>, <.>, `<newline>`, ␣and, ␣a, ␣of, ␣to, ␣is, ␣in |
| MMLU | ␣the, <,>, <.>, ␣of, ␣a, ␣to, ␣that, ␣and, ␣in, ␣was |
| AlpacaEval | ␣the, ␣a, ␣to, <,>, ␣of, ␣and, ␣I, <.>, ␣for, <?> |
| TQA | ␣the, <?>, ␣a, ␣you, ␣is, ␣of, ␣that, ␣to, ␣if, ␣in |

#### B.3.3 AVERAGE LID AND STANDARD DEVIATION OF PROMPTS WITH AND WITHOUT STOP WORDS AND PUNCTUATION

Table 35 shows the average LID and standard deviation of prompts with and without stop words and punctuation.

Table 35: Comparison of Average LID and Standard Deviation (SD) of Prompts with and without Stop Words and Punctuation

| Data Type | Dataset | With Stop Words and Punctuation | | Without Stop Words and Punctuation | |
|---|---|---|---|---|---|
| | | Avg. LID | SD | Avg. LID | SD |
| Benign | Orca | 9.77 | 2.52 | 6.08 | 1.91 |
| | MMLU | 9.23 | 2.86 | 5.64 | 1.76 |
| | AlpacaEval | 13.14 | 104.58 | 4.07 | 2.26 |
| | TQA | 6.29 | 3.40 | 3.93 | 0.65 |
| Adversarial | SAP | 12.10 | 0.55 | 7.27 | 0.46 |
| | DAN | 24.52 | 244.72 | 7.18 | 1.18 |
| | MWP | 8.98 | 1.44 | 4.14 | 1.28 |
| | GCG | 8.35 | 0.40 | 6.18 | 0.41 |

Table 36: Top 10 Most Common Nearest Neighbor Words for each dataset after removing stop words and punctuation. The symbol ⌴ represents a space preceding a word or punctuation.

| Dataset | Top 10 most common nearest neighbors |
|---|---|
| SAP | ⌴Remember, ⌴write, ⌴goal, ⌴mission, ⌴act, ⌴suicide, ⌴Use, ⌴phrases, ⌴use, ⌴refer |
| DAN | PT, ⌴D, G, AN, ⌴Chat, ⌴answer, ⌴AI, ⌴responses, ⌴response, ⌴respond |
| MathAttack | ⌴many, ⌴much, ⌴would, ⌴apples, ⌴money, ⌴20, ⌴sold, ⌴5, ⌴bought, ⌴day |
| GCG | use, ⌴mar, ⌴text, dt, end, ate, c, package, ized, ⌴t |
| Orca | ⌴answer, à, º, ⌴question, à¦, ⌴one, ⌴following, ⌴said, s, ⌴Answer |
| MMLU | ⌴mortgage, acre, ⌴state, ⌴contract, ⌴deed, ⌴would, ⌴question, ⌴statute, ⌴action, s |
| AlpacaEval | ⌴drinks, ⌴gathering, ⌴interested, ⌴give, ⌴time, ⌴home, br, ⌴trying, ⌴dishes, ⌴guests |
| TQA | ⌴say, ks, ⌴Oz, ⌴established, ⌴famous, ⌴primed, ⌴mirror, es, ⌴principle, ⌴power |

### B.3.4  TOP 10 MOST COMMON NEAREST NEIGHBOR WORDS AFTER REMOVING STOP WORDS AND PUNCTUATION

Table 36 shows the top 10 most common Nearest Neighbor words after removing stop words and punctuation. The experimental result demonstrates that word-level LID is insufficient for distinguishing between benign and adversarial prompts, even after the removal of stop words and punctuation.

## B.4 PERFORMANCE OF OTHER SOTA DEFENCES

To ensure a comprehensive evaluation, we organize this section into two parts. In Section B.4.1, we present results from running SOTA defences on our local environment, providing a consistent and fair comparison against CurvaLID. In Section B.4.2, we summarize the reported performances of other SOTA defences cited from their respective papers, offering a broader context across different LLMs.

### B.4.1 EVALUATION OF SOTA DEFENCES

In this subsection, we first present the results of evaluating five baseline defences—SmoothLLM (Robey et al., 2023), Self-Reminder (Xie et al., 2023), Intentionanalysis (Zhang et al., 2024a), In-Context Demonstration defence (ICD) (Wei et al., 2023), and RTT3d (Yung et al., 2024)—alongside our proposed CurvaLID. We evaluated these methods across four LLMs: Vicuna-7B-v1.1 (Chiang et al., 2023), Llama2-7B-Chat (Touvron et al., 2023), GPT-3.5 (Brown, 2020), and PaLM2 (Anil et al., 2023), and against seven different types of adversarial prompts: GCG (Zou et al., 2023), PAIR (Chao et al., 2023), DAN (Shen et al., 2023), AmpleGCG (Liao & Sun, 2024), SAP (Deng et al., 2023a), MathAttack (Zhou et al., 2024b), and RandomSearch (Andriushchenko et al., 2024). The experiment setting follows B.1.5 and the word embedding used for CurvaLID is RoBERTa. The experimental results, measured by ASR (%), are reported in Tables 37. CurvaLID consistently outperforms the baseline defences across most scenarios. We emphasize that all results are obtained through independent evaluation under a unified experimental setup.

Table 37: We compare CurvaLID with state-of-the-art defences, including SmoothLLM (Robey et al., 2023), Self-Reminder (Xie et al., 2023), Intentionanalysis (Zhang et al., 2024a), ICD (Wei et al., 2023), and RTT3d (Yung et al., 2024). The best results are **boldfaced**.

| LLM | defence | GCG | PAIR | DAN | AmpleGCG | SAP | MathAttack | RandomSearch |
|---|---|---|---|---|---|---|---|---|
| | No defence | 86.0 | 98.0 | 44.5 | 98.0 | 69.0 | 24.0 | 94.0 |
| **Vicuna-7B** | SmoothLLM | 5.5 | 52.0 | 13.0 | 4.2 | 44.6 | 22.0 | 48.5 |
| | Self-Reminder | 9.5 | 48.0 | 35.5 | 11.5 | 25.2 | 22.0 | 6.0 |
| | Intentionanalysis | **0.0** | 8.5 | 3.3 | **0.3** | 0.23 | 20.0 | **0.0** |
| | ICD | 0.2 | 5.2 | 40.4 | 0.9 | 32.8 | 22.0 | 0.2 |
| | RTT3d | 0.2 | 0.3 | 22.0 | 3.5 | 33.5 | 20.2 | 2.5 |
| | **CurvaLID** | **0.0** | **0.0** | **0.0** | 1.1 | **0.0** | **0.0** | **0.0** |
| | No defence | 12.5 | 19.0 | 2.0 | 81.0 | 9.5 | 11.7 | 90.0 |
| **LLaMA2-7B** | SmoothLLM | **0.0** | 11.0 | 0.2 | 0.2 | 1.2 | 11.2 | **0.0** |
| | Self-Reminder | **0.0** | 8.0 | 0.3 | **0.0** | **0.0** | 11.1 | **0.0** |
| | Intentionanalysis | **0.0** | 5.8 | 0.7 | **0.0** | **0.0** | 11.2 | **0.0** |
| | ICD | **0.0** | 2.7 | 0.8 | **0.0** | **0.0** | 10.8 | **0.0** |
| | RTT3d | 0.2 | 0.2 | 1.8 | 0.4 | 5.5 | 9.8 | 0.8 |
| | **CurvaLID** | **0.0** | **0.0** | **0.0** | **0.0** | **0.0** | **0.0** | **0.0** |
| | No defence | 12.0 | 48.0 | 6.33 | 82.0 | 0.9 | 10.5 | 73.0 |
| **GPT-3.5** | SmoothLLM | **0.0** | 4.9 | 0.3 | 0.7 | **0.0** | 10.9 | **0.0** |
| | Self-Reminder | **0.0** | 0.9 | 2.5 | 0.3 | 0.1 | 11.2 | **0.0** |
| | Intentionanalysis | **0.0** | 0.9 | 0.8 | 0.3 | **0.0** | 10.0 | **0.0** |
| | ICD | **0.0** | 0.7 | 0.9 | 0.4 | 0.4 | 9.9 | **0.0** |
| | RTT3d | 0.3 | 0.2 | 0.8 | 0.2 | 0.7 | 7.6 | **0.0** |
| | **CurvaLID** | **0.0** | **0.0** | **0.0** | **0.0** | **0.0** | **0.0** | **0.0** |
| | No defence | 14.9 | 98.0 | 49.7 | 88.9 | 55.1 | 18.9 | 91.9 |
| **PaLM2** | SmoothLLM | 5.5 | 38.7 | 6.7 | 7.2 | 41.2 | 9.8 | 45.3 |
| | Self-Reminder | 2.3 | 36.7 | 22.3 | 4.7 | 21.4 | 13.3 | 3.7 |
| | Intentionanalysis | **0.0** | 2.3 | 1.3 | 0.9 | **0.0** | 9.7 | **0.0** |
| | ICD | 0.1 | 4.9 | 34.2 | 0.2 | 33.9 | 9.3 | **0.0** |
| | RTT3d | 0.1 | 0.1 | 25.5 | 3.3 | 25.0 | 10.2 | 2.8 |
| | **CurvaLID** | **0.0** | **0.0** | **0.0** | **0.0** | **0.0** | **0.0** | **0.0** |

We also evaluated constrained SFT(Qi et al., 2025), using the fine-tuned Gemma-2-9B model released by the authors on GitHub and HuggingFace. The experimental results are shown in Table 38. We observe that constrained SFT is effective in mitigating attacks that rely on gibberish prefixes or suffixes, such as GCG and AmpleGCG, but fails to defend against social-engineering-based attacks like PAIR, DAN, and SAP. Most importantly, CurvaLID outperforms the constrained SFT defence across all adversarial attack types.

Table 38: Comparison of defences on Gemma-2-9B, measured by ASR (%).

| LLM | defence | GCG | PAIR | DAN | AmpleGCG | SAP | MathAttack | RandomSearch |
|---|---|---|---|---|---|---|---|---|
| | No defence | 90.2 | 23.8 | 80.0 | 81.3 | 44.5 | 22.8 | 94.5 |
| **Gemma-2-9B** | Constrained SFT | 22.5 | 28.5 | 83.5 | 29.2 | 78.8 | 22.0 | 90.0 |
| | **CurvaLID** | **0.0** | **0.0** | **0.0** | **2.1** | **0.0** | **0.0** | **0.0** |

### B.4.2 REPORTED PERFORMANCE OF EXISTING DEFENCES

We summarize the reported performances of other SOTA defences cited from their respective papers across different LLMs, focusing on their ability to reduce the ASR of adversarial prompts and compare this to CurvaLID's unified performance. CurvaLID outperforms the studied defences and maintains consistent performance across all LLMs. While we primarily compare four key defences in this subsection—SmoothLLM (Robey et al., 2023), Intentionanalysis (Zhang et al., 2024a), RTT3d (Yung et al., 2024), and LAT (Sheshadri et al., 2024)—which are considered SOTA or represent some of the most recent developments, we also experimented with other defences like Gradient Cuff (Hu et al., 2024), SELFDEFEND (Wang et al., 2024), SafeDecoding (Xu et al., 2024a), Circuit Breakers (Zou et al., 2024), Llama Guard (Inan et al., 2023), and perplexity-based filtering (Alon & Kamfonas, 2023).

Given the computational and time constraints associated with replicating results and testing across multiple LLMs, we have cited the performance figures for these defences from their respective papers. This approach ensures fairness, as different studies report varying results for these defences when replicating them against different adversarial prompts and across different models. Therefore, we rely on the original reported performances to provide a balanced and consistent comparison. Note that since the results for these defences are primarily based on English adversarial prompts in their respective papers, our analysis here is focused solely on English prompts.

defences based on input perturbation demonstrate mixed results depending on the LLM and the nature of the adversarial attack. For instance, Intentionanalysis reduces the ASR to between 0.03% and 8.34%, but it struggles with models like Vicuna-7B and MPT-30B-Chat, where the ASR for the SAP attack can reach nearly 20%. Similarly, while SmoothLLM can reduce the ASR to nearly 0% in various LLMs, it fails against PAIR attacks, showing ASRs of 46% in Vicuna-13B and 24% in GPT-4. RTT3d, as the first defence against MathAttack, managed to mitigate 40% of MathAttack in GPT4, but it failed to reduce the ASR to under 10%. LAT achieves near-zero ASR for models like Llama2-7B-Chat and Llama3-8B-Instruct. However, its reliance on a white-box setting and its testing on models with fewer than 10 billion parameters limit its broader applicability.

In the remainder of this subsection, we present the defensive performance of the SOTA defences. The performance figures for all defences are directly cited from their original papers due to the computational and time constraints involved in replicating results and testing across various LLMs. This approach ensures consistency and fairness in comparison, as the results reported by different studies often vary when replicating these defences on different adversarial prompts and models. By relying on the figures from the original sources, we aim to provide an accurate and balanced reflection of each defence's performance.

The following are the experimental settings and performances of the seven defences we studied. Note that it includes various LLMs, namely Vicuna (Chiang et al., 2023), Llama-2 (Touvron et al., 2023), Llama-3 (AI@Meta, 2024), GPT-3.5 (Brown, 2020), GPT-4 (Achiam et al., 2023), PaLM2 (Anil et al., 2023), Claude-1 (Anthropic, 2023a), Claude-2 (Anthropic, 2023b), ChatGLM-6B (Zeng et al., 2022), MPT-30B-Chat (Team, 2023), DeepSeek-67B-Chatand (Bi et al., 2024). It also includes various adversarial prompts, namely, GCG (Zou et al., 2023), PAIR (Chao et al., 2023), DAN (Shen et al., 2023), AmpleGCG (Liao & Sun, 2024), SAP (Deng et al., 2023a), MathAttack (Zhou et al., 2024b), RandomSearch (Andriushchenko et al., 2024), Prefill (Haizelabs, 2023), Many-Shot (Anil et al., 2024), AutoDAN (Liu et al., 2023), TAP (Mehrotra et al., 2023), Jailbroken (Wei et al., 2024), LRL (Yong et al., 2023), DrAttack (Li et al., 2024), Puzzler (Chang et al., 2024), MultiJail (Deng et al., 2023b), DeepInception (Li et al., 2023a), and Template (Yu et al., 2023).

**SmoothLLM** Detailed experimental settings are referred to (Robey et al., 2023). The experimental results are shown in Table 39.

Table 39: ASR comparison of different LLMs against adversarial prompts under SmoothLLM defence. Results are directly cited from the original paper. A dash (-) indicates that experiments were not conducted for this setting in the original paper.

| LLM | Adversarial Prompt | | | |
|---|---|---|---|---|
| | GCG | PAIR | RandomSearch | AmpleGCG |
| Vicuna-13B-v1.5 | 0.8 | 46 | 44 | 2 |
| Llama-2-7B-chat | 0.1 | 8 | 0 | 0 |
| GPT-3.5 | 0.8 | 2 | 0 | 0 |
| GPT-4 | 0.8 | 24 | 0 | 0 |
| PaLM-2 | 0.9 | - | - | - |
| Claude-1 | 0.3 | - | - | - |
| Claude-2 | 0.3 | - | - | - |

**Latent Adversarial Training** Detailed experimental settings are referred to (Sheshadri et al., 2024). The experimental results are shown in Table 40.

Table 40: ASR comparison of different LLMs against adversarial prompts under LAT defence. Results are directly cited from the original paper.

| LLM | Adversarial Prompt | | | | |
|---|---|---|---|---|---|
| | PAIR | Prefill | AutoPrompt | GCG | Many-Shot |
| Llama2-7B-chat | 0.025 | 0.029 | 0.006 | 0.007 | 0 |
| Llama3-8B-instruct | 0.0033 | 0.0068 | 0 | 0.009 | 0 |

**Gradient Cuff** Detailed experimental settings are referred to (Hu et al., 2024). The experimental results are shown in Table 41.

Table 41: ASR comparison of different LLMs against adversarial prompts under Gradient Cuff defence. Results are directly cited from the original paper.

| LLM | Adversarial Prompt | | | | | |
|---|---|---|---|---|---|---|
| | GCG | AutoDAN | PAIR | TAP | Base64 | LRL |
| Llama2-7B-chat | 0.012 | 0.158 | 0.23 | 0.05 | 0.198 | 0.054 |
| Vicuna-7B-v1.5 | 0.108 | 0.508 | 0.306 | 0.354 | 0 | 0.189 |

**Intentionanalysis** Detailed experimental settings are referred to (Zhang et al., 2024a). The experimental results are shown in Table 42.

Table 42: ASR comparison of different LLMs against adversarial prompts under Intentionanalysis defence. Results are directly cited from the original paper. A dash (-) indicates that experiments were not conducted for this setting in the original paper.

| LLM | Adversarial Prompt | | | | |
|---|---|---|---|---|---|
| | DAN | SAP200 | DeepInception | GCG | AutoDAN |
| ChatGLM-6B | 5.48 | 6.12 | 0 | 1 | 2 |
| LLaMA2-7B-Chat | 0.13 | 0 | 0 | 0 | 0 |
| Vicuna-7B-v1.1 | 3.42 | 0.31 | 0 | 0 | 10.5 |
| Vicuna-13B-v1.1 | 0.94 | 1.12 | 0 | 0 | 3.5 |
| MPT-30B-Chat | 5.38 | 19.2 | 4.78 | 4 | - |
| DeepSeek-67B-Chat | 3.78 | 1.56 | 7.57 | 2 | - |
| GPT-3.5 | 0.64 | 0 | 0 | 0 | - |

**SELFDEFEND** Detailed experimental settings are referred to (Wang et al., 2024). The experimental results are shown in Table 43.

Table 43: ASR comparison of different LLMs against adversarial prompts under SELFDEFEND defence. Results are directly cited from the original paper.

| LLM | Adversarial Prompt | | | | | | | |
|---|---|---|---|---|---|---|---|---|
| | DAN | GCG | AutoDAN | PAIR | TAP | DrAttack | Puzzler | MultiJail |
| GPT-3.5 | 0.007 | 0.18 | 0.31 | 0.29 | 0.02 | 0.71 | 0.22 | 0.203 |
| GPT-4 | 0.002 | 0 | 0.01 | 0.1 | 0.08 | 0.04 | 0.26 | 0.012 |

**RTT3d** Detailed experimental settings are referred to (Yung et al., 2024). The experimental results are shown in Table 44.

Table 44: ASR comparison of different LLMs against adversarial prompts under RTT3d defence. Results are directly cited from the original paper. A dash (-) indicates that experiments were not conducted for this setting in the original paper.

| LLM | Adversarial Prompt | | | |
|---|---|---|---|---|
| | PAIR | GCG | SAP | MathAttack |
| GPT-3.5 | - | - | 0.06 | 9.8 |
| GPT-4 | 0.265 | - | - | - |
| Llama-2-13B-Chat | 0.043 | 0.17 | - | - |
| Vicuna-13B-v1.5 | 0.26 | 0.15 | - | - |
| PaLM-2 | 0.13 | - | - | - |

**SafeDecoding** Detailed experimental settings are referred to (Xu et al., 2024a). The experimental results are shown in Table 45.

Table 45: ASR comparison of different LLMs against adversarial prompts under SafeDecoding defence. Results are directly cited from the original paper.

| LLM | Adversarial Prompt | | | | | |
|---|---|---|---|---|---|---|
| | GCG | AutoDAN | PAIR | DeepInception | SAP30 | Template |
| Vicuna-7B | 0.04 | 0 | 0.04 | 0 | 0.09 | 0.05 |
| Llama2-7B-Chat | 0 | 0 | 0.04 | 0 | 0 | 0 |

**Circuit Breakers** Detailed experimental settings are referred to (Zou et al., 2024). The experimental results are shown in Table 46.

Table 46: Comparison between CurvaLID and Circuit Breakers on LLaMA-3-8B. Results are reported in terms of ASR (%).

| LLM | defence | Adversarial Prompt | | | | | |
|---|---|---|---|---|---|---|---|
| | | GCG | PAIR | DAN | AmpleGCG | SAP | RandomSearch |
| LLaMA-3-8B | Circuit Breakers | 2 | 3.33 | 0 | 2.5 | 0 | 0 |
| | CurvaLID | 0 | 0 | 0 | 2.5 | 0 | 0 |

**Llama Guard** Detailed experimental settings are referred to (Inan et al., 2023). The experimental results are shown in Table 47.

Table 47: Comparison of LLaMA Guard 3 and CurvaLID. Results are reported in terms of ASR (%).

| defence | PAIR | DAN | SAP | AutoDAN | GCG | AmpleGCG | MMLU | AlpacaEval |
|---|---|---|---|---|---|---|---|---|
| LLaMA Guard 3 | 0 | 0 | 0 | 0 | 0 | 0 | 0.09 | 0.01 |
| CurvaLID | 0 | 0 | 0 | 0 | 0 | 0.025 | 0 | 0.02 |

**Perplexity-based filtering** Detailed experimental settings are referred to (Alon & Kamfonas, 2023). The experimental results are shown in Table 48.

Table 48: Comparison of Perplexity Filtering and CurvaLID. Results are reported in terms of ASR (%).

| defence | GCG | AmpleGCG | DAN | AutoDAN | SAP | PAIR | Persuasive Attack |
|---|---|---|---|---|---|---|---|
| Perplexity Filtering | 0 | 0 | 0.475 | 0.38 | 1 | 0.624 | 0.76 |
| CurvaLID | 0 | 0.015 | 0 | 0 | 0 | 0 | 0 |

## C   LLM USAGE

This research directly concerns LLMs, and all experiments necessarily involved their usage. In addition, we used LLMs in a limited capacity to aid and polish the writing of this paper.

