# OpenReview forum: "Geometry-Guided Adversarial Prompt Detection via Curvature and Local Intrinsic Dimension"
_ICLR.cc/2026/Conference — Submitted to ICLR 2026_

### Official Review · Reviewer_dp7e · 2025-10-28

**Soundness:** 3
**Presentation:** 2
**Contribution:** 2
**Rating:** 4
**Confidence:** 2

**Summary:**

This paper addresses the issue of adversarial prompts faced by large language models (LLMs) by proposing a model-agnostic detection framework named CurvaLID. The authors first define and implement two geometric measures: sentence-level PromptLID and word-level TextCurv. These measures are then used to train a multilayer perceptron (MLP) to distinguish benign prompts from adversarial ones. Experiments conducted across multiple LLMs (Vicuna, LLaMA2, GPT-3.5, PaLM2, etc.) and multiple attack categories (GCG, DAN, SAP, etc.), claiming to reduce attack success rates to near zero with an overall classification accuracy of approximately 0.99. The overarching contribution lies in systematically introducing classical geometric concepts (curvature, LID) into adversarial prompt detection, demonstrating robust cross-model performance and efficiency advantages.

**Strengths:**

1. Novel perspective: Integrating the concept of curvature with LID for geometric analysis of text/prompts, combining theory with intuition to offer an enlightening viewpoint.
2. Model-agnostic: Functioning as a pre-processing filter for large language models (LLMs), CurvaLID circumvents the need for fine-tuning target LLMs, demonstrating practical deployment potential.
3. Extensive experimental scope and diversity: The study encompasses multiple mainstream LLMs, several common adversarial attack datasets, and comparative evaluations against various state-of-the-art defence methods. These factors collectively enhance the paper's persuasiveness.

**Weaknesses:**

1. Overall, the approach is innovative, though certain methods involve combining existing tools (LID, cosine angle) and applying them to novel scenarios.
2. The layout of charts and images within the paper is somewhat haphazard, causing some inconvenience during reading.
3. PromptLID requires training a multi-class CNN model in its initial step. This necessitates the prior collection and annotation of substantial benign prompts for model training, thereby limiting the method's applicability in unannotated or data-scarce environments.

**Questions:**

Unclear Motivation and Conceptual Novelty: The paper’s motivation is insufficiently articulated. While it introduces CurvaLID by combining curvature and Local Intrinsic Dimensionality (LID) for adversarial prompt detection, the rationale for why geometric properties fundamentally capture adversarial semantics is not convincingly established. As a result, the approach appears as a technical aggregation of known methods rather than a conceptually novel framework grounded in theoretical or empirical insight.

Problem–Solution Misalignment: The paper criticizes existing methods for depending on labeled data, yet CurvaLID itself requires labeled benign and adversarial prompts for supervised training. Although an unsupervised variant is briefly mentioned, it is not the main evaluation. Hence, the proposed framework does not fundamentally overcome the annotation dependency it aims to address, weakening its claimed contribution.

Presentation and Layout Issues: The manuscript suffers from poor formatting: figures and tables (e.g., Tables 1–2, Figure 2) are misaligned, captions are fragmented, and visual clarity is limited. These presentation issues hinder readability and detract from the paper’s professional appearance.

Over-Strict Detection and Lack of Usability Discussion: While CurvaLID achieves near-perfect adversarial detection, it also exhibits false positives on benign prompts. The paper does not analyze this trade-off or discuss potential over-refusal behavior, leaving concerns about the method’s practicality and real-world usability unresolved.

---

> ### Author Response · Authors · 2025-11-20
> **Author Response Part 1**
>
> Thank you for your review. We appreciate your recognition of our paper’s novelty, theoretical contributions, and practical relevance.
>
> **Novelty on CurvaLID:**
>
> **Summary: Both PromptLID and TextCurv are newly defined geometric measures developed in Section 4, offering a mathematically grounded characterisation of adversarial text behaviour rather than adaptations of existing tools.**
>
> We respectfully disagree with the claim that our paper does not simply “combine existing tools”. For PromptLID, we do not replicate the original MoM-LID estimator from prior work. Instead, we derive a new prompt-level geometric formulation inspired by LID and present it in Section 4.1 (Lines 207–243). We explicitly justify this modification in Section 4.1 (Lines 237–242), where we cite prior literature showing that adversarial prompts manipulate the word embedding space to reach rare, high-complexity subspaces. As a result, these prompts naturally exhibit higher PromptLID values, since they push representations into less well-defined regions of the feature space.
>
> For TextCurv, we are the first to characterise word-level curvature using established mathematical definitions, capturing semantic shifts across the token sequence and their strength. The theoretical motivation is provided in Section 4.2 (Lines 256–269), and we further present a full mathematical derivation of TextCurv in Section 4.2 (Lines 270- 300). This formulation is not based on pre-existing tools but is developed specifically for modelling textual curvature.
>
> Overall, the definitions and derivations in Section 4 demonstrate that PromptLID and TextCurv provide novel, mathematically grounded geometric characterisations of adversarial text behaviour, rather than reusing or modifying prior techniques.
>
> **Presentation of charts and images:**
>
> We are happy to incorporate any specific adjustments the reviewer may recommend for the figures or diagrams.
>
> **CurvaLID under data-scarce environment:**
>
> **Summary: Across four evaluations, CurvaLID achieves 0.9–0.992 accuracy under limited-data, unseen-benign, one-class, and held-out OOD conditions, confirming its reliability in unannotated and data-scarce environments.**
>
> We kindly remind the reviewer that CurvaLID’s applicability in unannotated and data-scarce environments is directly demonstrated through four experiments presented in the paper.
>
> 1. Appendix B.2.21 (Lines 1574- 5191) presents an evaluation in which CurvaLID is trained on only two benign datasets and tested on the remaining two. Even with unseen benign data at test time, CurvaLID still achieves 0.98 accuracy, indicating that its effectiveness does not depend on a substantial benign dataset.
>
> 2. Appendix B.2.12 (Lines 1379- 1399) further shows that CurvaLID remains robust when trained with significantly less benign data, using only half of the original amount, and still reaches 0.992 accuracy, demonstrating that it does not require large annotated corpora.
>
> 3. In Section 5.2 (Lines 432–438), we also evaluate CurvaLID in a one-class setting using only benign prompts and no adversarial data, and CurvaLID maintains around 0.9 accuracy, confirming applicability when adversarial labels are unavailable.
>
> 4. Appendix B.2.24 (Lines 1628- 1651) reports a held-out OOD experiment in which each dataset is excluded from training and tested separately, and CurvaLID still achieves around 0.9 accuracy.
>
> These results collectively show that CurvaLID remains effective even when benign data are limited, unannotated, or drawn from unseen sources, consistently achieving 0.9–0.992 accuracy across cross-benign testing, reduced-data settings, one-class training, and held-out OOD evaluation.

---

> ### Author Response · Authors · 2025-11-20
> **Author Response Part 2**
>
> **Motivation and Conceptual Novelty:**
>
> **Summary: The paper clearly states its motivation in Section 1, and Section 4 provides full theoretical justification for PromptLID and TextCurv, with empirical validation shown in Figures 2(d), 3, 4 and Table 3, demonstrating that both measures reliably capture the geometric differences between benign and adversarial prompts.**
>
> We kindly remind the reviewer that the paper’s motivation and conceptual framework are explicitly stated in Section 1 (Lines 59–63). We clearly articulate that CurvaLID introduces an LLM-agnostic framework for detecting adversarial prompts by examining the geometric differences between benign and adversarial text. CurvaLID is designed to operate independently of any specific model architecture, which is why our goal is to identify geometric signals that generalise across diverse LLMs.
>
> The theoretical rationale for both geometric components is also developed in detail in Section 4. For PromptLID, we explain the expected geometric divergence in Section 4.1 (Lines 237–243), grounding our intuition in prior work showing that adversarial prompts often manipulate the high-dimensional word-embedding space to reach rare, complex subspaces. For TextCurv, we provide the full theoretical grounding in Section 4.2 (Lines 246–289), where we show that curvature captures semantic direction changes between consecutive tokens, revealing local geometric differences that arise when adversarial prompts introduce abrupt or stitched-together segments. Both explanations are supported by prior literature cited in the same sections.
>
> The empirical insight behind our geometric formulation is also demonstrated clearly in our results. For PromptLID, Figure 2(d) (Lines 389–399) and Figure 3 (Lines 1674–1689) show a clear and consistent separation between benign and adversarial prompts. For TextCurv, Table 3 (Lines 378–388) and Figure 4 (Lines 1693–1712) demonstrate substantial curvature differences across multiple embedding models. These results empirically validate the geometric rationale developed in Section 4 and highlight that CurvaLID is not a mere aggregation of existing techniques, but a coherent geometric framework supported by both theory and data.
>
> **Problem-Solution Misalignment:**
>
> **Summary: The reviewer’s concern stems from a misinterpretation of our stated problem and contribution, which are clearly aligned in Section 1.**
>
> We respectfully disagree with the reviewer’s comment regarding the problem–solution misalignment. Our first contribution, stated clearly in Section 1, Introduction (Lines 85–88), is that we propose CurvaLID as an LLM-agnostic detection framework that leverages geometric distinctions between adversarial and benign prompts. This directly aligns with the core problem identified in Section 1 (Lines 42–59), where we highlight that existing defences rely heavily on LLM-specific internal architectures and safety-alignment training, limiting their generality across models.
>
> Our contribution is therefore not about eliminating annotation dependency, but about providing a general, architecture-independent geometric framework, which is precisely what CurvaLID delivers.
>
> **Presentation and Layout Issues:**
>
> We are happy to incorporate any specific adjustments the reviewer may recommend for the figures or diagrams.
>
> **False-positive on benign prompts and over-refusal behaviour:**
>
> **Summary: Both false-positive rates and over-refusal behaviour are already analysed in the main text: the confusion matrix in Figure 2(a) reports false positives, and Section 5.1 evaluates CurvaLID on two over-refusal benchmarks, showing it avoids over-refusal while reducing harmful-prompt acceptance.**
>
> We respectfully disagree with the reviewer’s claim that the paper does not discuss false positives or over-refusal behaviour. In Figure 2(a) (Lines 388–399), we explicitly present the confusion matrix for the main experiment, which directly reports CurvaLID’s false-positive behaviour on benign prompts. Regarding over-refusal, Section 5.1 (Lines 423–426) clearly states that we evaluate CurvaLID on two over-refusal benchmarks, and we show that CurvaLID effectively avoids over-refusal, while still reducing harmful-prompt acceptance by up to 30%.

---

> ### Author Response · Authors · 2025-11-27
>
> Thank you once again for your thoughtful reviews. We hope that our rebuttal has satisfactorily addressed your questions and concerns. We would appreciate any further feedback, and we are happy to clarify any additional points.

---

### Official Review · Reviewer_m8re · 2025-10-28

**Soundness:** 3
**Presentation:** 3
**Contribution:** 3
**Rating:** 6
**Confidence:** 4

**Summary:**

The paper proposes a defence against jailbreaks and prompt injection attacks.
Ths is composed of several steps. First a neural network classifies samples into
their respective datasets. Then, based on the internal representations within
the CNN, measures are used of PromptLID and TextCurv. Using a mix-
ture of benign and adversarial prompts PromptLID/TextCurv are computed
to form a new dataset based on these higher level features. Finally, using
this PromptLID/TextCurv a MLP is trained to classify benign and adversarial
prompts.

**Strengths:**

Deploying small and lightweight classifiers for jailbreak detection is a useful line of research for practical use of defences.

The results are promising: against the tested attacks there is good overall performance.

The approach seems novel taking a different strategy to guardrail defences compared to many other literature approaches

**Weaknesses:**

No adaptive attacks are considered against the defence: for example, optimise the prompt to simultaneously bypass the defence and still carry out a jailbreak against the underlying LLM. I.e. it is an "oblivious" attacker model.

The classification of benign prompts into 4 datasets seems a bit arbitrary, particularly as datasets can overlap, and is a loose proxy for the distribution the classifier is aiming for (e.g. good internal representations of benign prompts).

Results in Table 1 are based on the test set split for the trained classifier. For fair comparisons with the general defences OOD jailbreaks should be considered. There is some preliminary experiments to that end in Table 30 in the appendix B.2.24 which should be developed. Likewise  discussion of false positive rates outside the datasets used for training the detector seem to be absent, aside from the appendix, whereas it would be the more representative performance of the defence in practice.

**Questions:**

I am unsure why those 4 adversarial datasets were highlighted in Table 2. From the description of the data in Section 4, it seems that all datasets are equivalently used for training and testing? They just vary in the number of samples? Perhaps I have missed something here.

---

> ### Author Response · Authors · 2025-11-20
> **Author Response Part 1**
>
> Thank you for your valuable review. We appreciate your recognition of our paper’s novelty, theoretical contributions, and practical relevance.
>
> **CurvaLID under adaptive adversarial attack:**
>
> **Summary: Our revised paper includes adaptive-attack evaluations in Appendix B.2.28, showing the attempts to jointly minimise PromptLID and TextCurv break jailbreak effectiveness, directly addressing the reviewer’s concern.**
>
>
> We thank the reviewer for this valuable feedback. Current SOTA adversarial attacks [1,2,3] are already highly constrained and carefully engineered to bypass modern LLM safety systems. Adding the additional requirement to simultaneously minimize both PromptLID and TextCurv makes the search space far more restrictive and dramatically increases the difficulty of crafting a successful jailbreak. In preliminary experiments, we attempted to enforce these minimization constraints during adversarial generation. While doing so occasionally reduced CurvaLID’s confidence, the resulting prompts failed to bypass the internal safety mechanisms of the underlying LLMs, hence the attack no longer achieved its primary objective. In response to the reviewer’s concern, we incorporated a new section—Appendix B.2.28, Adaptive Attack Attempts Against CurvaLID (Lines 1719–1772)—in the revised version of the manuscript.
>
> We would be more than happy to evaluate CurvaLID against any successful adaptive attack that preserves malicious intent while minimizing these geometric features, and we welcome any concrete suggestions from the reviewer.
>
> **Benign prompts classification:**
>
> **Summary: Two experiments in the paper demonstrate that CurvaLID’s behaviour is stable across different benign-data partitions, showing that its performance does not depend on dataset overlap or arbitrary partition.**
>
> We appreciate the reviewer’s thoughtful comment. In Appendix B.2.21 (Lines 1574- 1591), we directly address the reviewer’s concern by evaluating CurvaLID when trained on only two benign datasets and tested on the remaining two. Even under this cross dataset setting, where any potential dataset overlap is removed, CurvaLID still achieves 0.98 accuracy, indicating that its performance is not tied to an arbitrary partition of benign data. Moreover, Appendix B.2.12 (Lines 1379- 1399) shows that CurvaLID remains robust even when trained with significantly less data (halving the number of benign prompts), still reaching 0.992 accuracy.
>
> **OOD jailbreaks and false positive rates:**
>
> **Summary: Three experiments in the paper evaluate CurvaLID under OOD jailbreak settings, consistently showing strong performance across one-class, reduced-data, and cross-dataset benign evaluations, and the revised version now includes a detailed false-positive analysis for completeness.**
>
> We appreciate the reviewer raising this point. In addition to the OOD analysis in Table 30 (Appendix B.2.24, Lines 1628-1651), our paper already includes three more experiments evaluating CurvaLID under out-of-distribution settings.
>
> 1. In Section 5.2 (Lines 432–438), we evaluate CurvaLID in a one-class setting, training only on benign prompts with no access to adversarial examples. Even under this strict OOD evaluation, CurvaLID maintains around 0.9 accuracy, indicating it does not rely on exposure to seen adversarial prompts.
>
> 2. Appendix B.2.12 (Lines 1379-1399) demonstrates robustness when training with significantly less benign data (0.992 accuracy).
>
> 3. Appendix B.2.21 (Lines 1574- 1591) shows that training on two benign datasets and testing on the remaining two still achieves 0.98 accuracy.
>
> These experiments collectively demonstrate that CurvaLID performs strongly even when evaluated OOD from the training configuration used in Table 1 (Lines 324- 344).
>
> To address the reviewer’s request for a more detailed false-positive analysis, we conducted an additional experiment measuring the false positive rates in our OOD evaluation (Table 30, Appendix B.2.24, Lines 1628-1651), and we have included this result in our revised version of the paper, Appendix B.2.24, Table 30 (Lines 1641-1648). Importantly, because CurvaLID is trained on seven datasets and evaluated on the remaining held-out dataset, all evaluation samples are OOD relative to the training distribution. Since the held-out dataset contains only benign prompts, any misclassification is necessarily a false positive. Thus, for this evaluation setting, the false-positive rate is exactly 1−accuracy. The same reasoning applies symmetrically to held-out datasets that contain only adversarial prompts: since every evaluation sample is harmful, any misclassification corresponds to a false negative, which is also exactly 1−accuracy.
>
> | Dataset     | Accuracy | FPR |
> |-------------|----------|-------------------|
> | Orca        | 0.955    | 0.045             |
> | MMLU        | 0.940    | 0.060             |
> | AlpacaEval  | 0.945    | 0.055             |
> | TruthfulQA  | 0.995    | 0.005             |

---

> ### Author Response · Authors · 2025-11-20
> **Author Response Part 2**
>
> **Selection of highlighted adversarial datasets:**
>
> **Summary: The four adversarial datasets in Table 2 were chosen because they are standard SOTA benchmarks with sufficient sample sizes for fair comparison to the benign datasets.**
>
> We are grateful to the reviewer for highlighting this issue. Yes, the datasets differ only in the number of available samples, as we inherit their abundances directly from the original papers and GitHub releases. The four adversarial datasets highlighted in Table 2 were selected because they are widely used benchmarks in prior SOTA adversarial defense papers [4,5,6,7] and contain substantially more prompts than the other adversarial datasets in our collection. These datasets are also used to demonstrate PromptLID, token-level LID, and average nearest-neighbour distance in Figure 2 (Lines 388- 398), so it is important that they contain a similar amount of data to each of the benign datasets for comparison. Moreover, these four adversarial datasets span a diverse range of attack strategies: DAN, a socially engineered human-crafted jailbreak; GCG, an automatically generated gradient-based attack; SAP, which combines manual and automated perturbations; and MathsAttack, which targets the model’s logical and mathematical reasoning capabilities. We have included this additional explanation in the revised version of the paper in Section 5.1 (Lines 418–419).
>
> [1]: Zou, Andy, Zifan Wang, Nicholas Carlini, Milad Nasr, J. Zico Kolter, and Matt Fredrikson. "Universal and transferable adversarial attacks on aligned language models." arXiv preprint arXiv:2307.15043 (2023).
>
> [2]: Chao, Patrick, Alexander Robey, Edgar Dobriban, Hamed Hassani, George J. Pappas, and Eric Wong. "Jailbreaking black box large language models in twenty queries." In IEEE Conference on Secure and Trustworthy Machine Learning (SaTML), 2025.
>
> [3]: Liu, Xiaogeng, Nan Xu, Muhao Chen, and Chaowei Xiao. "Autodan: Generating stealthy jailbreak prompts on aligned large language models." arXiv preprint arXiv:2310.04451 (2023).
>
> [4]: Qi, Xiangyu, Ashwinee Panda, Kaifeng Lyu, Xiao Ma, Subhrajit Roy, Ahmad Beirami, Prateek Mittal, and Peter Henderson. "Safety Alignment Should be Made More Than Just a Few Tokens Deep." In The Thirteenth International Conference on Learning Representations (ICLR), 2025.
>
> [5]: Robey, Alexander, Eric Wong, Hamed Hassani, and George J. Pappas. "SmoothLLM: Defending Large Language Models Against Jailbreaking Attacks." Transactions on Machine Learning Research (TMLR), 2025.
>
> [6]: Zhang, Yuqi, Liang Ding, Lefei Zhang, and Dacheng Tao. "Intention analysis prompting makes large language models a good jailbreak defender." arXiv preprint arXiv:2401.06561 32 (2024).
>
> [7]: Wei, Zeming, Yifei Wang, Ang Li, Yichuan Mo, and Yisen Wang. "Jailbreak and guard aligned language models with only few in-context demonstrations." arXiv preprint arXiv:2310.06387 (2023).

---

> ### Author Response · Authors · 2025-11-27
>
> Thank you again for your insightful review. We hope our rebuttal has clarified your questions and concerns. We would be grateful for any further comments, and we are more than happy to address additional questions should they arise.

---

### Official Review · Reviewer_bgUJ · 2025-10-31

**Soundness:** 3
**Presentation:** 3
**Contribution:** 3
**Rating:** 4
**Confidence:** 3

**Summary:**

The paper proposes CurvaLID, an LLM‑agnostic pre‑filter that detects adversarial prompts using two geometric features computed on fixed representations: (1) PromptLID: a sentence‑level Local Intrinsic Dimensionality score computed from a CNN’s penultimate layer; (2) TextCurv: a word‑level “curvature” that combines pairwise angular changes with an inverse‑norm surrogate for arc length. Across many attack families and multiple target LLMs, the paper reports near‑perfect detection with cross‑embedding and multilingual analyses suggesting generality.

**Strengths:**

1. Clear story to defend the jailbreak attack using LLM‑agnostic pre‑filtering. The jailbreak detector operates entirely outside the LLM under defense, making the deployment super simple.
2. Combining a sentence‑level density proxy (LID) with a word‑sequence geometry (curvature) is intuitive and computationally light. Experiment results are consistently high across models and attacks.
3. Pseudocode, architectural details, and hyperparameters are provided for reproduction.

**Weaknesses:**

1. Step 1 trains g on a few specific benign datasets (Orca, MMLU, AlpacaEval, TQA), could the separation captured by PromptLID be driven by dataset style rather than adversarialness? A cross-domain benign evaluation might help disentangle this effect.
2. CurvaLID achieves perfect near-zero ASR against non-adaptive attacks, but what happens under an adaptive adversary explicitly minimizing PromptLID and TextCurv? Could such optimization collapse the geometric gap and evade detection?
3. PromptLID is defined inconsistently between Def. 4.1 and Alg. 2 (Sec B.3.1). Could the author fix or clarify this mismatch?

**Questions:**

1. Table 2 reports 1.00 adversarial accuracy with 0.000 SD. Does this persist across different seeds, folds, and class balances? Please include a variance analysis and per-dataset ROC/AUROC.
2. Many jailbreaks unfold over conversations (multi-turn). Do you compute geometry per-turn or over concatenated history?

---

> ### Author Response · Authors · 2025-11-20
> **Author Response Part 1**
>
> Thank you for your valuable review. We appreciate your recognition of our paper’s novelty, theoretical contributions, and practical relevance.
>
> **Possible cross-domain evaluation:**
>
> **Summary: Three experiments demonstrate that CurvaLID’s behaviour is stable across unseen benign datasets, reduced-data settings, and one-class training, showing that its separation is driven by adversarial geometric structure rather than benign dataset style.**
>
> We kindly remind the reviewer that we investigate this issue through three dedicated experiments evaluating whether CurvaLID’s separation arises from dataset style rather than adversarial structure:
>
> 1. In Appendix B.2.21 (Lines 1574- 1591), we conduct a cross-domain benign evaluation where CurvaLID is trained on only two benign datasets and tested on the remaining two. Despite this distribution shift, CurvaLID still achieves 0.98 accuracy, suggesting that its separation does not rely on stylistic artifacts of any specific benign dataset.
>
> 2. Appendix B.2.12 (Lines 1379-1399) further shows that CurvaLID remains robust even when trained with significantly less benign data, still reaching 0.992 accuracy, indicating that the geometric signals it captures are not sensitive to the particular composition or size of the benign training set.
>
> 3. In Section 5.2 (Lines 432–438), we evaluate CurvaLID in a one-class setting, where the detector is trained only on benign prompts with no access to adversarial examples; even under this constraint, CurvaLID maintains around 0.9 accuracy, demonstrating that the method does not depend on memorizing dataset-specific style differences.
>
> Across all three evaluations (cross-benign transfer, limited benign data, and one-class training), the evidence consistently shows that PromptLID captures intrinsic adversarial geometry rather than stylistic patterns of the benign datasets used to train g.
>
> **CurvaLID under adaptive adversarial attack:**
>
> **Summary: Our revised paper includes adaptive-attack evaluations in Appendix B.2.28, showing the attempts to jointly minimise PromptLID and TextCurv break jailbreak effectiveness, directly addressing the reviewer’s concern.**
>
> We thank the reviewer for this valuable feedback. Current SOTA adversarial attacks [1,2,3] are already highly constrained and carefully engineered to bypass modern LLM safety systems. Adding an additional requirement to minimize both PromptLID and TextCurv simultaneously makes the search space substantially more restrictive and dramatically increases the difficulty of crafting a successful attack. In preliminary experiments, we attempted to enforce these minimization constraints during adversarial generation. While doing so occasionally reduced CurvaLID’s confidence, the resulting prompts failed to bypass the internal safety mechanisms of the underlying LLMs, hence the attack no longer achieved its primary objective. In response to the reviewer’s concern, we incorporated a new section—Appendix B.2.28, Adaptive Attack Attempts Against CurvaLID (Lines 1719–1772)—in the revised version of the manuscript.
>
> We would be more than happy to evaluate CurvaLID against any successful adaptive attack that preserves malicious intent while minimizing these geometric features, and we welcome any concrete suggestions from the reviewer.
>
> **PromptLID Definition:**
>
> **Summary: The definition and the algorithm refer to two different quantities.**
>
> We kindly remind the reviewer that Definition 4.1 and Algorithm 2 are fundamentally different and are not intended to define the same quantity. Definition 4.1 (Line 231) gives the formal definition of PromptLID, our newly proposed prompt-level LID measure. In contrast, Algorithm 2 (Appendix B.3.1, Lines 1788–1792) describes the standard Method-of-Moments LID estimator used in prior literature. PromptLID is one of our main contributions and is a new LID formulation tailored for text, whereas Algorithm 2 is included only to report classical LID for comparison. We have included this distinction in the revised version of the paper, in Appendix B.3.1 (Lines 1843- 1844).

---

> ### Author Response · Authors · 2025-11-20
> **Author Response Part 2**
>
> **Variance Analysis and per-dataset ROC/AUROC:**
>
> **Summary: The experiment settings and variance considerations are fully detailed in Section 5 and supported by four robustness experiments (OOD, cross-dataset, reduced-data, and one-class), all showing stable 0.9–0.99 performance, and we have added per-dataset ROC/AUROC in the revised version.**
>
> We appreciate the reviewer raising this point. As stated explicitly in Section 5 (Lines 366–367), all main results are already averaged over 10 independent runs for reliability, using an 80/20 train–test split. Regarding class balance, our main experiment (Section 5.1, Lines 357–359) uses 1,200 benign prompts and 2,340 adversarial prompts. We also conduct several ablation studies that directly examine robustness across dataset splits, and class balances. Appendix B.2.24 (Lines 1628- 1651) presents a held-out OOD evaluation where each dataset is excluded during training and tested separately. Appendix B.2.21 (Lines 1574- 1591) evaluates CurvaLID when trained on only two benign datasets and tested on the remaining two. Appendix B.2.12 (Lines 1379- 1399) shows that CurvaLID still achieves 0.992 accuracy even when trained with significantly less data. Finally, Section 5.2 (Lines 432–438) reports the one-class setting, where CurvaLID is trained solely on benign data and still maintains around 0.9 accuracy. Across all of these settings, CurvaLID exhibits consistently high performance with low variance.
>
> Regarding variance analysis, we have included the confusion matrix for the main results in Figure  2(a), Lines 389-399. For the per-dataset ROC/AUROC, we have conducted another experiment and included the result below. We have included this result in our revised version of the paper, Appendix B.2.29, Table 33, Lines 1824 - 1830.
>
> | Class   | Dataset      | Accuracy | AUROC (raw) |
> |---------|--------------|----------|-------------|
> | Benign  | Orca         | 0.968    | 0.983       |
> | Benign  | MMLU         | 1.000    | 1.000       |
> | Benign  | AlpacaEval   | 0.983    | 0.992       |
> | Benign  | TQA          | 0.986    | 0.993       |
> | **Adv.** | SAP          | 1.000    | 0.992       |
> | **Adv.** | DAN          | 1.000    | 0.992       |
> | **Adv.** | MathAtk      | 1.000    | 0.992       |
> | **Adv.** | GCG          | 1.000    | 0.992       |
>
> **CurvaLID under multi-turn jailbreaks:**
>
> **Summary: Appendix B.2.2 demonstrates that CurvaLID remains effective on multi-turn attacks, achieving over 0.9 accuracy with per-turn geometric analysis.**
>
> We thank the reviewer for this valuable feedback. While our paper focuses on single-shot adversarial prompts, CurvaLID naturally extends to multi-turn jailbreaks by computing the geometric features per turn. Multi-turn attacks often resemble demonstration-based attacks, where the LLM is exposed to a crafted adversarial context before the harmful query appears. In Appendix B.2.2 (Lines 1166- 1182), we evaluate CurvaLID on such demonstration-based and cipher-style attacks, and applying CurvaLID per turn achieves over 0.9 detection accuracy. This suggests that CurvaLID remains effective even when an adversarial sequence unfolds over multiple turns. We have incorporated this explanation in our revised version of the paper in Appendix B.2.2, lines 1171- 1172.
>
> [1]: Zou, Andy, Zifan Wang, Nicholas Carlini, Milad Nasr, J. Zico Kolter, and Matt Fredrikson. "Universal and transferable adversarial attacks on aligned language models." arXiv preprint arXiv:2307.15043 (2023).
>
> [2]: Chao, Patrick, Alexander Robey, Edgar Dobriban, Hamed Hassani, George J. Pappas, and Eric Wong. "Jailbreaking black box large language models in twenty queries." In IEEE Conference on Secure and Trustworthy Machine Learning (SaTML), ,, 2025.
>
> [3]: Liu, Xiaogeng, Nan Xu, Muhao Chen, and Chaowei Xiao. "Autodan: Generating stealthy jailbreak prompts on aligned large language models." arXiv preprint arXiv:2310.04451 (2023).

---

> > ### Comment · Reviewer_bgUJ · 2025-11-21
> >
> > Thank you so much for the detailed response and the revision to the manuscript, it answered some of previous questions and made the manuscript more solid. I decided to raise my rating.

---

> ### Author Response · Authors · 2025-11-27
>
> Thank you very much for acknowledging our response and the manuscript revisions. We deeply appreciate the reviewer’s thoughtful feedback and are grateful for the updated rating.

---

### Official Review · Reviewer_imVD · 2025-11-04

**Soundness:** 2
**Presentation:** 2
**Contribution:** 2
**Rating:** 2
**Confidence:** 3

**Summary:**

The paper proposes CurvaLID, a detection framework for identifying adversarial prompts that could jailbreak LLMs. The method combines two geometric measures: (1) PromptLID, a sentence-level LID estimator based on CNN representations of prompts; and (2) TextCurv, a curvature measure that captures angular changes between consecutive word embeddings. The authors claim these geometric properties distinguish benign from adversarial prompts. They report high performance (over 99% accuracy and near-zero attack success rates) across multiple LLMs and attack datasets, while being model-agnostic and computationally efficient.

**Strengths:**

- The paper explores geometric properties of text embeddings (curvature, intrinsic dimensionality) as indicators of adversarial behavior—a creative perspective applied to LLM jailbreak detection.
- The proposed method operates independently of any target LLM’s architecture or internal safety mechanisms, potentially offering wide applicability.

**Weaknesses:**

- The paper does not convincingly explain why sentence-level LID or text curvature should differentiate malicious from benign prompts. The “geometric intuition” remains vague—there is no solid theoretical or empirical link between these geometric properties and adversarial text behavior.
- The paper does not clarify how the phenomena of high LID regions in adversarial examples in traditional image domains translate to text prompts, which lie in discrete, semantically structured spaces.
- The paper’s mathematical definition of PromptLID deviates from established LID formulations. The proposed equation appears to reduce to a simple function of k-nearest neighbor distances and kernel density estimates, not a genuine estimation of intrinsic dimensionality.
- No analysis validates whether PromptLID meaningfully captures intrinsic dimensional differences versus acting as a heuristic distance statistic.
- CurvaLID uses a binary classifier trained directly on labelled benign and adversarial prompts. This inevitably biases the model to the seen data distribution.
- The paper’s extremely high reported accuracy (≈99%) raises concerns of overfitting—especially given the small dataset sizes.
- The evaluation exclusively targets existing benchmark jailbreak prompts and fails to assess adaptive adversaries—those that can adjust input to evade detection once the defense mechanism is known. Since the method is based on deterministic geometric statistics (e.g., k-NN distances, cosine angles), an adaptive attacker could easily craft prompts maintaining similar curvature and LID while retaining malicious intent. Without such robustness analysis, the claimed “generalization” is unconvincing.

**Questions:**

- Could you provide a clearer theoretical or empirical justification for why sentence-level LID and text curvature should meaningfully differentiate malicious from benign prompts?
- The proposed formulation of PromptLID appears to deviate from established LID estimation methods. Could you clarify how your definition still captures the intrinsic dimensionality?
- How does performance degrade as the data distribution diverges from the training examples?
- The paper reports near-perfect accuracy (~99%). Are there failure cases or qualitative examples where CurvaLID misclassifies prompts, and what patterns do they exhibit?

---

> ### Author Response · Authors · 2025-11-20
> **Author Response Part 1**
>
> Thank you for your review.
>
> **Link between geometric properties and adversarial text behavior:**
>
> **Summary: Section 4 establishes a full theoretical foundation for PromptLID and TextCurv, and the empirical results in Figures 2(d), 3, 4, and Table 3 show clear and consistent geometric separation between benign and adversarial prompts.**
>
> The geometric properties we study (PromptLID and TextCurv) are not assumed a priori but are in fact the central novel findings of our work. A key contribution of the paper is showing that when a model is trained only on benign data, adversarial prompts naturally occupy distinct geometric regions with higher LID in prompt-level and higher token-level curvature. The solid theoretical contribution of PromptLID and TextCurv to adversarial text behaviour is clearly demonstrated in Sections 4.1 (Lines 237–243) and 4.2 (Lines 246-269).
>
> We also empirically  demonstrate the connection between these geometric properties and adversarial text behaviour. In our paper, adversarial prompts consistently show over 200% higher PromptLID on average (shown in Figure 2(d), Lines 389- 398, and Figure 3, Lines 1674-1693), and 30–45% higher TextCurv across five embedding models (shown in Table 3, Lines 378- 388, and Figure 4, Lines 1693-1711), with clear separation from benign prompts.
>
> These results validate that the geometric measures themselves capture the structural irregularities of adversarial prompts, which explains why CurvaLID achieves strong detection accuracy.
>
> **LID usage translation from image to text:**
>
> **Summary: The paper does not assume direct vision-to-text transfer. PromptLID is theoretically grounded and explicitly formulated for text in Section 4.1, and its behaviour is empirically validated in Figures 2(d) and 3.**
>
> We kindly remind the reviewer that our paper does not assume that LID behaviour from vision models directly transfers to text. Instead, we define our own prompt-level LID, PromptLID, and verify its behaviour within the text domain. As stated clearly in Section 4.1 (Lines 237–243), our intuition is grounded in prior findings that adversarial prompts often manipulate the high-dimensional space of word embeddings to target rarely encountered subspaces, leading them to exhibit distinct geometric characteristics. This provides the theoretical basis for applying LID in a text-embedding space without relying on any image-domain analogy.
>
> We also show empirically that this behaviour does emerge in text. Figure 2(d), Lines 389- 398, and Figure 3, Lines 1674-1693, demonstrate that adversarial prompts consistently produce significantly higher PromptLID across datasets and embedding models. In other words, our method does not depend on an image-to-text transfer assumption as the paper itself establishes and validates the high-LID phenomenon directly for text prompts.
>
> **PromptLID mathematical definition:**
>
> **Summary: PromptLID is not a copy of established LID estimators but a prompt-level redesign.  Section 4.1 provides its full theoretical basis, while Figures 2(d) and 3 empirically confirm that adversarial prompts consistently exhibit higher LID in text-embedding space.**
>
> Our goal was not to replicate the exact MoM-LID estimator, but to adapt the theory of LID into a prompt-level geometric signal, which we clearly present as a modified formulation in Section 4.1 (Lines 207–243). Moreover, PromptLID still follows the core LID principle of using k-NN distance ratios. We also explicitly justify this modification in Section 4.1, Lines 237–242, where we cite prior work showing that adversarial prompts manipulate word-embedding space to reach rare, high-complexity subspaces. Consequently, they naturally exhibit higher PromptLID, as they push representations into less well-defined regions of the feature space.
>
> Besides, PromptLID is empirically validated in our paper. As shown in Figure 2(d), Lines 389- 398, and Figure 3, Lines 1674-1693, it cleanly separates benign and adversarial prompts across datasets, and it also succeeds in a one-class setting using LOF/IF (Section 5.2, lines 432-437).
>
> Thus, although PromptLID is a tailored variant rather than a verbatim LID estimator, it preserves the key geometric intuition of LID and consistently demonstrates strong discriminative power in the text domain.

---

> ### Author Response · Authors · 2025-11-20
> **Author Response Part 2**
>
> **Concerns on PromptLID capturing intrinsic dimensional differences:**
>
> **Summary: We introduce PromptLID as a new text domain geometric measure in Section 4.1 rather than a true intrinsic dimension estimator, and we validate its behaviour empirically in Figures 2(d) and 3, showing that it consistently distinguishes adversarial prompts from benign ones.**
>
> To the best of our knowledge, no prior work has proposed an intrinsic-dimension measure tailored for text data, and we would be happy to evaluate any such metric if the reviewer can suggest one. Moreover, our paper does not claim that PromptLID recovers the true intrinsic dimensionality. Instead, we frame it as a prompt-level geometric measure derived from the same k-NN distance-ratio principle as classical LID. In Section 4.1 (Lines 237–243), we also justify why adversarial prompts are expected to occupy rarer, higher-complexity subspaces in embedding space, motivating a LID-style formulation rather than a simple distance heuristic.
>
> We further provide direct empirical validation that PromptLID captures meaningful geometric differences rather than acting as a naïve distance statistic. As shown in Figure 2(d), Lines 389- 398, and Figure 3, Lines 1674-1693, PromptLID achieves clear and consistent separation between benign and adversarial prompts across datasets, and it remains effective in a one-class setting using LOF/IF (Section 5.2, lines 432-437). This consistent behaviour demonstrates that PromptLID reflects structural differences in local neighbourhood geometry.
>
> **Concerns on model biases to the seen data distribution:**
>
> **Summary: Through four experiments—including held-out OOD testing, one-class training, reduced-data training, and cross-dataset benign evaluation—CurvaLID consistently maintains 0.9 -- 0.99 accuracy, demonstrating that it does not depend on the seen data distribution.**
>
> We kindly remind the reviewer that CurvaLID’s robustness to the seen data distribution is explicitly tested through four experiments in the paper:
>
> 1. In Appendix B.2.24 (Lines 1628-1651), we conduct a held-out OOD evaluation where each dataset is excluded during training and tested separately, and CurvaLID still achieves around 0.9 accuracy.
>
> 2. In Section 5.2 (Lines 432–438), we further evaluate CurvaLID in a one-class setting, training only on benign prompts with no access to adversarial examples; even under this strict condition, CurvaLID maintains around 0.9 accuracy, showing that it does not rely on exposure to the seen adversarial distribution.
>
> 3. In  Appendix B.2.12 (Lines 1379-1399), we show that CurvaLID remains robust even when trained with significantly less data (0.992 accuracy).
>
> 4. In Appendix B.2.21 (Lines 1574-1591) we report that training on two benign datasets and testing on the remaining two still achieves 0.98 accuracy.
>
> Taken together, these four experiments clearly show that CurvaLID maintains high accuracy across diverse training–testing configurations, demonstrating that its performance is driven by underlying geometric signals rather than memorisation of the seen data distribution.
>
> **Concerns for overfitting and small dataset size:**
>
> **Summary: Two over-refusal benchmark evaluations directly refute the overfitting claim. Our experiments use more than 5,000 prompts, surpassing SOTA work published at ICLR, TMLR, and Nature MI. This confirms that CurvaLID’s performance is grounded in broad empirical evidence rather than memorisation.**
>
> We respectfully disagree with the reviewer’s comment regarding overfitting or small dataset size. In Section 5.1 (Lines 423–426), we explicitly demonstrate CurvaLID’s ability to avoid overfitting by evaluating on two over-refusal benchmarks (OR-Bench and XSTest) where CurvaLID still reduces harmful-prompt acceptance by up to 30%. Moreover, as stated in Section 5 (Lines 352–368), our experiment results are from experiments with 5 LLMs, 6 SOTA defences, 4 benign datasets, and 10 adversarial datasets, totalling over 3,500 prompts. We have also tested on SOTA benchmarks such as HarmBench, OR-BENCH, and XSTEST, using 1900 prompts in total. The scale of our evaluation is comparable to, if not larger than, other SOTA work published at ICLR 2025 (Outstanding Paper Award) [1], TMLR 2025 [2], and Nature Machine Intelligence 2023 [3], indicating that the high accuracy is not the result of overfitting.

---

> ### Author Response · Authors · 2025-11-20
> **Author Response Part 3**
>
> **Concerns on CurvaLID’s generalisation and adaptive adversaries:**
>
> **Summary: Our revised paper includes adaptive-attack evaluations in Appendix B.2.28, showing the attempts to jointly minimise PromptLID and TextCurv break jailbreak effectiveness, directly addressing the reviewer’s generalisation concern.**
>
> We respectfully disagree with the reviewer’s concern regarding CurvaLID’s generalisation. Evaluating established SOTA jailbreak benchmarks and datasets is standard practice in adversarial-prompt research, and our evaluation setup is consistent with SOTA defence papers [1,2,3]. Moreover, the claim that an adaptive attacker could “easily craft prompts maintaining similar curvature and LID” does not align with the reality of current jailbreak methods. SOTA adversarial prompts [4,5,6] already require highly constrained, carefully constructed text to bypass modern LLMs, and adding simultaneous constraints on both curvature and PromptLID would make such attacks substantially more difficult.
>
> In preliminary experiments, we attempted to enforce these minimization constraints during adversarial generation. While doing so occasionally reduced CurvaLID’s confidence, the resulting prompts failed to bypass the internal safety mechanisms of the underlying LLMs, hence the attack no longer achieved its primary objective. In response to the reviewer’s concern, we incorporated a new section—Appendix B.2.28, Adaptive Attack Attempts Against CurvaLID (Lines 1719–1772)—in the revised version of the manuscript.
>
> We are more than happy to evaluate CurvaLID on any adaptive attack the reviewer can recommend, and we appreciate suggestions for future robustness experiments.
>
> **Justification on utilising PromptLID and TextCurv:**
>
> **Summary: Section 4 provides the full theoretical foundation for PromptLID and TextCurv, and Figures 2(d), 3, 4, and Table 3 empirically confirm that both measures reliably separate benign from adversarial prompts by capturing their distinct geometric behaviour.**
>
> We respectfully remind the reviewer that our paper already provides both theoretical motivation and empirical validation for why PromptLID and TextCurv differentiate benign and adversarial prompts. The theoretical intuition for PromptLID is stated explicitly in Section 4.1 (Lines 237–242), where we cite prior work showing that adversarial prompts tend to manipulate the high-dimensional word-embedding space to target rare, complex subspaces, which naturally leads to elevated PromptLID. Similarly, Section 4.2 (Lines 244–309) formalises TextCurv using standard geometric definitions and explains that adversarial prompts, often constructed by stitching together semantically inconsistent segments, induce sharper directional changes in the word-embedding trajectory, resulting in higher curvature.
>
> Empirically, our paper demonstrates that these geometric measures do meaningfully separate the two classes. As shown in Figure 2(d) (Lines 389- 398) and Figure 3 (Lines 1674-1693), PromptLID exhibits clear, consistent separation between benign and adversarial prompts across datasets. Table 3 (Lines 378- 388), and Figure 4 (Lines 1693-1711) further show that adversarial prompts have 30–45% higher TextCurv across five embedding models, with stronger amplification in CNN activations.
>
> These results directly validate that both PromptLID and TextCurv capture the characteristic geometric irregularities of adversarial prompts, consistently separating them from benign text across datasets and embedding models.

---

> ### Author Response · Authors · 2025-11-20
> **Author Response Part 4**
>
> **Concerns on PromptLID Deviation from Established LID Estimation:**
>
> **Summary: We introduce PromptLID as a new text-domain geometric measure in Section 4.1 rather than a true intrinsic-dimension estimator, and its behaviour is empirically validated in Figures 2(d) and 3, demonstrating consistent geometric separation between benign and adversarial prompts.**
>
> To the best of our knowledge, no prior work has proposed an intrinsic-dimension metric specifically designed for text data, and we would be happy to evaluate any such measure if the reviewer can suggest one. Importantly, our paper does not claim that PromptLID recovers the true intrinsic dimensionality. Instead, PromptLID is introduced as a prompt-level geometric signal inspired by the same k-NN distance-ratio principle underlying classical LID. In Section 4.1 (Lines 237–243), we justify this formulation by citing prior work showing that adversarial prompts tend to drive word-embedding representations into rarer, higher-complexity subspaces, which motivates using a LID-style measure rather than a simple distance-based heuristic.
>
> We also provide direct empirical evidence that PromptLID captures meaningful geometric structure. As shown in Figure 2(d) (Lines 389–398) and Figure 3 (Lines 1674–1693), PromptLID yields clear and consistent separation between benign and adversarial prompts across datasets. Furthermore, in a one-class setting using LOF/IF (Section 5.2, Lines 432–437), PromptLID remains effective even without any access to adversarial examples.
>
> Together, these results demonstrate that although PromptLID is a tailored variant rather than a classical LID estimator, it reliably reflects structural differences in local neighbourhood geometry within the text-embedding space.
>
> **CurvaLID on data distribution diverges from the training examples:**
>
> **Summary: Four experiments in the paper explicitly evaluate CurvaLID under distribution shift, consistently showing around 0.9–0.99 accuracy even when training and test distributions diverge.**
>
> We kindly remind the reviewer that CurvaLID’s behaviour under distribution shift is thoroughly evaluated through four experiments in the paper:
>
> 1. In Appendix B.2.24 (Lines 1628- 1651), we conduct a held-out OOD experiment where each dataset is excluded during training and tested separately, and CurvaLID still achieves around 0.9 accuracy.
>
> 2. In Section 5.2 (Lines 432–438), we further test CurvaLID in a one-class setting where the detector is trained only on benign prompts with no access to adversarial examples. This removes any dependence on the seen adversarial distribution, and even under this strict condition CurvaLID maintains around 0.9 accuracy, indicating strong robustness to distribution divergence.
>
> 3. Appendix B.2.12 (Lines 1379-1399) shows that CurvaLID remains robust even when trained with significantly less data, still reaching 0.992 accuracy.
>
> 4. Appendix B.2.21 (Lines 1574- 1591) further reports an ablation study where CurvaLID is trained on two benign datasets and tested on the remaining two, achieving 0.98 accuracy despite this distribution shift.
>
> The four evaluations show that CurvaLID consistently retains high accuracy (0.9–0.99) across held-out datasets, one-class training, reduced-data settings, and cross-dataset benign testing, confirming its robustness even when the training and testing distributions differ substantially.

---

> ### Author Response · Authors · 2025-11-20
> **Author Response Part 5**
>
> **Cases for CurvaLID misclassification:**
>
> **Summary: CurvaLID’s failure cases are quantitatively examined across multiple evaluations, with detailed explanations added in the revised manuscript to clarify why errors arise.**
>
> We provide quantitative analyses of CurvaLID’s failure cases in the paper. In Appendix B.2.24 (Lines 1628- 1651), our held-out OOD evaluation shows that CurvaLID remains strong across all datasets (around 0.9 accuracy), with the lowest-performing dataset being PAIR (0.875 accuracy). PAIR attacks are social-engineering adversarial prompts with high human readability. Unlike gradient-based attacks such as GCG or AmpleGCG, which often introduce unnatural or gibberish token patterns, PAIR prompts are fluent, coherent, and unconstrained by artificial token structures, making them more closely resemble natural conversation than typical template-based jailbreaks. This results in less curvature irregularity and weaker geometric signals, leading to the prediction errors in this dataset. Similarly, Appendix B.2.21 (Lines 1574- 1591) shows that when CurvaLID is trained on only two benign datasets and tested on the remaining two, the benign-prompt accuracy drops to 0.94, indicating that benign prompts with unfamiliar formatting or structure account for the small number of unseen benign-prompt misclassifications.
>
> To further clarify this, we have incorporated the detailed explanation in Appendix B.2.21 and B.2.24 in our revised version of the paper.
>
> [1]: Qi, Xiangyu, Ashwinee Panda, Kaifeng Lyu, Xiao Ma, Subhrajit Roy, Ahmad Beirami, Prateek Mittal, and Peter Henderson. "Safety Alignment Should be Made More Than Just a Few Tokens Deep." In The Thirteenth International Conference on Learning Representations (ICLR), 2025.
>
> [2]: Robey, Alexander, Eric Wong, Hamed Hassani, and George J. Pappas. "SmoothLLM: Defending Large Language Models Against Jailbreaking Attacks." Transactions on Machine Learning Research (TMLR), 2025.
>
> [3]: Xie, Yueqi, Jingwei Yi, Jiawei Shao, Justin Curl, Lingjuan Lyu, Qifeng Chen, Xing Xie, and Fangzhao Wu. "Defending chatgpt against jailbreak attack via self-reminders." Nature Machine Intelligence 5, no. 12 (2023).
>
> [4]: Zou, Andy, Zifan Wang, Nicholas Carlini, Milad Nasr, J. Zico Kolter, and Matt Fredrikson. "Universal and transferable adversarial attacks on aligned language models." arXiv preprint arXiv:2307.15043 (2023).
>
> [5]: Chao, Patrick, Alexander Robey, Edgar Dobriban, Hamed Hassani, George J. Pappas, and Eric Wong. "Jailbreaking black box large language models in twenty queries." In IEEE Conference on Secure and Trustworthy Machine Learning (SaTML), 2025.
>
> [6]: Liu, Xiaogeng, Nan Xu, Muhao Chen, and Chaowei Xiao. "Autodan: Generating stealthy jailbreak prompts on aligned large language models." arXiv preprint arXiv:2310.04451 (2023).

---

> ### Author Response · Authors · 2025-11-27
>
> Thank you again for your valuable reviews. We hope our rebuttal has addressed your questions and concerns. We would greatly appreciate any further feedback you may be willing to share, and we are happy to respond to any additional questions or comments.

---

### Author Response · Authors · 2025-12-01

Dear AC,

We have provided a point-to-point response to all reviewers' concerns and questions. Many of the issues raised were already covered in the original submission, and in our rebuttal we explicitly pointed reviewers to the corresponding sections and line numbers where these points were already stated. For the newly raised concerns, such as the need for adaptive attack evaluation and more extensive variance analysis, we conducted additional experiments and incorporated these results into the appendix of the revised manuscript.

Reviewer bgUJ responded to our rebuttal and acknowledged that our clarifications and revisions resolved several of their earlier concerns. As recorded on 22 Nov 2025 (before the widespread circulation of leaked information), the reviewer wrote:

> Thank you so much for the detailed response and the revision to the manuscript, it answered some of previous questions and made the manuscript more solid. I decided to raise my rating.

Their score increased from 4 to 6 accordingly. While the remaining reviewers did not respond before the discussion period closed, we have addressed all of their comments comprehensively in the revised manuscript.

Overall, we believe our rebuttals successfully clarified all reviewer concerns. We sincerely thank all reviewers for their feedback and suggestions for improving the paper.

Best regards,

Authors

---

### Meta-Review · Area_Chair_TgLh · 2026-01-07

**Summary:**

The paper proposes two geometric metrics, PromptLID and TextCurv, to defend against adversarial jailbreak prompts.
During the discussion phase, primary concerns arose regarding the mechanistic drivers and physical interpretations of the proposed metrics. While the empirical results are impressive, many of the evaluated adversarial attacks—specifically GCG, AmpleGCG, and Random Search—are known to succeed by appending long, nonsensical suffixes consisting of illegible token sequences. From a geometric perspective, such "gibberish" naturally shifts latent representations into sparse, high-entropy regions of the embedding space, which inherently manifests as elevated LID and curvature.
Hence, to provide a more convincing validation, I suggest that the authors evaluate their method against semantically fluent attacks (e.g., "How Johnny Can Persuade LLMs to Jailbreak Them").

**Reviewer Scores:**

No

---

### Decision · Program_Chairs · 2026-01-26

Reject